# Comprehensive and clinically accurate head and neck cancer organs-at-risk delineation on a multi-institutional study

Xianghua Ye[1,15], Dazhou Guo[2,15], Jia Ge[1,15], Senxiang Yan[1], Yi Xin[3], Yuchen Song[1], Yongheng Yan[1], Bing-shen Huang[4], Tsung-Min Hung[4], Zhuotun Zhu[5], Ling Peng[6], Yanping Ren[7], Rui Liu[8], Gong Zhang[9], Mengyuan Mao[10], Xiaohua Chen[11], Zhongjie Lu[1], Wenxiang Li[1], Yuzhen Chen[4], Lingyun Huang[3], Jing Xiao[3], Adam P. Harrison[12], Le Lu[2], Chien-Yu Lin[4,13] ✉, Dakai Jin[2] ✉ & Tsung-Ying Ho[14] ✉

Accurate organ-at-risk (OAR) segmentation is critical to reduce radiotherapy complications. Consensus guidelines recommend delineating over 40 OARs in the head-and-neck (H&N). However, prohibitive labor costs cause most institutions to delineate a substantially smaller subset of OARs, neglecting the dose distributions of other OARs. Here, we present an automated and highly effective stratified OAR segmentation (SOARS) system using deep learning that precisely delineates a comprehensive set of 42 H&N OARs. We train SOARS using 176 patients from an internal institution and independently evaluate it on 1327 external patients across six different institutions. It consistently outperforms other state-of-the-art methods by at least 3–5% in Dice score for each institutional evaluation (up to 36% relative distance error reduction). Crucially, multi-user studies demonstrate that 98% of SOARS predictions need only minor or no revisions to achieve clinical acceptance (reducing workloads by 90%). Moreover, segmentation and dosimetric accuracy are within or smaller than the inter-user variation.

Head and neck (H&N) cancer is one of the most common cancers worldwide[1]. Radiation therapy (RT) is an important and effective treatment for H&N cancer[2]. In RT, the radiation dose to normal anatomical structures, i.e., organs at risk (OARs), needs to be limited to reduce post-treatment complications, such as dry mouth, swallowing difficulties, visual damage, and cognitive decline[3–6]. This requirement demands accurate OAR delineation on the planning computed tomography (pCT) images used to configure the radiation dosage treatment. Recent consensus guidelines recommend a set of more than 40 OARs in the H&N region[7]. Nevertheless, precise manual delineation of this quantity of OARs is an overwhelmingly demanding task that requires great clinical expertise and time efforts, e.g., >3 h for 24 OARs[8]. Due to the factors of patient overload and shortage of experienced physicians, long patient waiting times and/or undesirably inaccurate RT delineations are more common than necessary, reducing the treatment efficacy and safety[9]. To shorten time expenses, many institutions choose a simplified (sometimes overly simplified) OAR protocol by contouring a small subset of OARs (e.g., only the OARs closest to the tumor). Dosimetric information cannot be recorded for non-contoured OARs, although it is clinically important to track for analysis of post-treatment side effects[10]. Moreover, because clinicians often follow the institution-specific OAR contouring style, manual delineation is easily prone to large inter-observer variations leading to differences/discrepancies in dose parameters potentially impacting the treatment outcome[7]. Therefore, automatic and accurate segmentation of a comprehensive set of H&N OARs is of great clinical benefit in this context.

OARs are spatially densely distributed in the H&N region and often have complex anatomical shapes, large size variations, and low CT contrasts. Conventional atlas-based methods have been extensively explored previously[11–15], but significant amounts of editing efforts were found to be unavoidable[8,16]. Atlas-based methods heavily rely on the accuracy and reliability of deformable image registration, which can be very challenging due to OARs' large shape variations, normal tissue removal, tumor growth, and image acquisition differences. Volumetric deformable registration methods often take many minutes or even hours to compute.

Deep learning approaches have shown substantial improvements in improving segmentation accuracy and efficiency as compared to atlas-based methods[17]. After early patch-based representation[18], fully convolutional network is the dominant formulation on segmentation[19–22] or adopting a segmentation-by-detection strategy[23,24] when the number of considered OARs is often fewer than or around 20. With a greater number of OARs needed to be segmented, deep network optimization may become increasingly difficult. From an early preliminary version of this work[25], we introduced a stratified deep learning framework to segment a comprehensive set of H&N OARs by balancing the OARs' intrinsic spatial and appearance complexity with adaptive neural network architectures. The proposed system, stratified organ at risk segmentation (SOARS), divides OARs into three levels, i.e., anchor, mid-level, and small & hard (S&H) according to their complexity. Anchor OARs are high in intensity contrast and low in inter-user variability and can be segmented first to provide informative location references for the following harder categories. Mid-level OARs are low in contrast but not inordinately small. We use anchor-level predictions as additional input to guide the mid-level OAR segmentation. S&H OARs are very small in size or very poor in contrast. Hence, we use a detection by segmentation strategy to better manage the unbalanced class distributions across the entire volume. Besides this processing stratification, we further deploy another stratification by using neural architecture search (NAS) to automatically determine the optimal network architecture for each OAR category since it is unlikely the same network architecture suits all categories equally. We specifically formulate this structure learning problem as differentiable NAS[26,27], allowing automatic selection across 2D, 3D, or Pseudo-3D (P3D) convolutions with kernel sizes of 3 or 5 pixels at each convolutional block.

SOARS segments a large number (42) of OARs with quantitatively leading performance in a single institution cross-validation evaluation[25], but essential questions remain unclear regarding its clinical applicability and generality: (1) does SOARS generalize well into a large-scale multi-institutional evaluation?; (2) how much manual editing effort is required before the predicted OARs can be considered as clinically accepted?; (3) how well does the segmentation accuracy of SOARS compare towards inter-user variation?; and more critically, (4) what are the dosimetric variations brought by OAR differences in the downstream RT planning stage? To adequately address these questions, we first enhance SOARS by replacing the segmentation backbone of P-HNN[28] with UNet[29] and conduct the NAS optimization based on the UNet architecture. Then, we extensively evaluate SOARS on an external set of 1327 unseen H&N cancer patients from six institutions (one internal and five external). Finally, using 50 randomly selected external patients (from two clinical sites), we further conducted three subjective user studies: (1) physician's assessment of the revision effort and time spent when editing on predicted OARs; (2) a comparison of contouring accuracy between SOARS and the inter-user variation; and (3) in the intensity modulated RT (IMRT) planning, a dosimetric accuracy comparison using different OAR contours (SOARS, SOARS + physician editing, and physician's manually labeling).

## Results

### Datasets for training and evaluation

In this multi-institutional retrospective study, we collected, in total, 1503 H&N cancer patients (each with a pCT scan and who received RT as their primary treatment) to develop and evaluate the performance of SOARS. Besides the pCT scans, MRI scans (if available) and other clinical information were also provided to physicians as references during their manual OAR delineation procedure. Radiologists were also consulted when encountering difficult cases, such as tumors very close to the OARs. Patients were collected from Chang Gung Memorial Hospital (CGMH), First Affiliated Hospital of Xi'an Jiaotong University (FAH-XJU), and First Affiliated Hospital of Zhejiang University (FAH-ZU), Gansu Provincial Hospital (GPH), Huadong Hospital Affiliated of Fudan University (HHA-FU), Southern Medical University (SMU). Detailed patient characteristics in each institution are shown in Table 1 and image scanning parameters in each institution are listed in Supplementary Table 1.

**Training-validation dataset.** First, we created a training-validation dataset to develop SOARS using 176 patients from CGMH between 2015 and 2018 (internal training dataset). Each patient had 42 OARs manually delineated by senior physicians (board-certified radiation oncologists specialized in HN cancer treatment) according to the consensus guideline[7] or delineation methods[30,31] recommended by the guideline[7]. Among these OARs, several subdivisions of brain structures were considered, because studies have reported the radiotherapy-induced fatigue, short-term memory loss, and cognition change associated with the volume of scatter dose to these brain substructures[32–36]. Note that a senior physician in our study is not only required to have experience in the head & neck specialty for at least 10 years with 100–300 annually treated patients but also is very familiar with and follows the delineation consensus guidelines[7] in their clinical practice with high fidelity. Based on the OAR statistical shape, CT appearance and location characteristics (confirmed by the physicians), 42 OARs are divided into the following three categories. Anchor OARs: brainstem, cerebellum, eye (left and right), mandible (left and right), spinal cord, and temporomandibular joint (TMJoint, left and right). Mid-level OARs: brachial plexus (left and right), basal ganglia (left and right), constrictor muscle (inferior, middle, and superior), esophagus, glottic and supraglottic larynx (GSL), glottic area, oral cavity, parotid (left and right), submandibular gland (SMG, left and right), temporal lobe (left and right), thyroid (left and right). S&H OARs: cochlea (left and right), hypothalamus, inner ear (left and right), lacrimal gland (left and right), lens (left and right), optic nerve (left and right), optic chiasm, pineal gland, and pituitary. These 42 OARs represent one of the most comprehensive H&N OAR sets and can serve as a superset when testing/evaluating patients in other institutions. We divided this dataset into two subgroups: 80% to train and validate the segmentation model and 20% to evaluate the ablation performance. Detailed data split protocols for the NAS training, and the ablation evaluation are reported in the supplementary materials. The ablation performance of SOARS is depicted in Table 2.

**Independent internal testing dataset.** Next, for independent evaluation, we collected 326 patients from CGMH between 2014 and 2020 as another internal testing dataset besides the training-validation. OAR labels in this cohort were extracted from those generated during the clinical RT contouring process that senior physicians examined and confirmed. Depending on the H&N cancer types or tumor locations, a range of 18–42 OAR contours were generally available for each patient in this cohort.

**Table 1 | Subject characteristics**

| Characteristics | Train/validation CGMH (n = 176) | Internal testing CGMH (n = 326) | External testing FAH-XJU (n = 82) | External testing FAH-ZU (n = 447) | External testing GPH (n = 50) | External testing HHA-FU (n = 195) | External testing SMU (n = 227) |
|---|---|---|---|---|---|---|---|
| **Sex** | | | | | | | |
| Male | 160 (91%) | 284 (87%) | 65 (79%) | 321 (72%) | 33 (66%) | 145 (75%) | 161 (71%) |
| Female | 16 (9%) | 42 (13%) | 17 (21%) | 126 (28%) | 17 (34%) | 50 (25%) | 66 (29%) |
| Diagnostic age | 54 [48–61] | 54 [49–62] | 57 [49–66] | 57 [50–65] | 58 [49–70] | 56 [47–65] | 50 [42–57] |
| **Tumor site** | | | | | | | |
| Nasopharynx | 7 (4%) | 90 (28%) | 16 (19%) | 349 (78%) | 2 (4%) | 94 (48%) | 199 (88%) |
| Oropharynx | 140 (80%) | 86 (26%) | 20 (24%) | 26 (6%) | — | 2 (1%) | 9 (4%) |
| Hypopharynx | 16 (9%) | 115 (35%) | — | 16 (4%) | — | 8 (4%) | 3 (1%) |
| Larynx | 2 (1%) | 12 (4%) | 38 (47%) | 11 (2%) | 9 (18%) | 25 (13%) | 4 (2%) |
| Oral Cavity | 9 (5%) | 15 (5%) | 9 (10%) | 39 (9%) | 3 (6%) | 2 (1%) | 5 (2%) |
| Salivary gland | — | — | — | — | 4 (8%) | 4 (2%) | 3 (1%) |
| Others | 2 (1%) | 8 (2%) | — | 6 (1%) | 32 (64%) | 60 (31%) | 4 (2%) |
| **Clinical T-stage** | | | | | | | |
| cT1 | 23 (13%) | 50 (15%) | 10 (12%) | 55 (12%) | 12 (24%) | 14 (7%) | 20 (9%) |
| cT2 | 64 (36%) | 82 (25%) | 33 (41%) | 181 (41%) | 18 (36%) | 64 (33%) | 54 (24%) |
| cT3 | 42 (24%) | 81 (25%) | 25 (30%) | 122 (27%) | 12 (24%) | 35 (18%) | 101 (44%) |
| cT4 | 47 (27%) | 113 (35%) | 14 (17%) | 89 (20%) | 8 (16%) | 82 (42%) | 52 (23%) |
| OAR types annotated | 42 | 42 | 13 | 13 | 17 | 13 | 25 |

*CGMH* Chang Gung Memorial Hospital, *FAH-XJU* First Affiliated Hospital of Xi'an Jiaotong University, *FAH-ZU* First Affiliated Hospital of Zhejiang University, *GPH* Gansu Provincial Hospital, *HHA-FU* Huadong Hospital Affiliated of Fudan University, *SMU* Southern Medical University.
Note: others of tumor sites include tumors located in the brain, nasal cavity, or lymph node metastasis.

**Table 2 | Quantitative results of the ablation evaluation using the validation set of the training-validation dataset**

| | Anchor OARs | | | Mid-level OARs | | | S&H OARs | | | All OARs | | |
|---|---|---|---|---|---|---|---|---|---|---|---|---|
| | DSC | HD | ASD | DSC | HD | ASD | DSC | HD | ASD | DSC | HD | ASD |
| Baseline nnUNet[39] | 84.3% | 12.4 | 1.0 | 71.4% | 18.0 | 2.0 | 58.3% | 4.7 | 1.1 | 70.4% | 12.7 | 1.4 |
| nnUNet + PS | 86.7% | 6.4 | 0.9 | 72.6% | 11.4 | 1.9 | 73.7% | 4.6 | 0.7 | 76.1% | 8.2 | 1.3 |
| nnUNet+PS+NAS | **87.4%** | **5.4** | **0.8** | **74.2%** | **10.4** | **1.7** | **76.2%** | **3.5** | **0.6** | **77.8%** | **7.2** | **1.2** |

Note: PS, NAS represent processing stratification and neural architecture search, respectively. The unit for Hausdorff distance (HD) and average surface distance (ASD) is in mm. The best performance scores are highlighted in bold font.

**Multi-institutional external testing dataset.** For quantitative external evaluation, 1001 patients were collected from five different institutions located in various areas of mainland China between 2014 and 2020 (external testing dataset). Each patient is accompanied by the clinical RT treatment OAR contours, ranging from 13 to 25 OARs, depending on their institutional-specific RT protocols. Two steps of examinations were conducted to ensure the accuracy and consistency of reference OAR contours among different institutions. First, senior physicians of each institution first examined and edited the clinical OAR contours of the data from their own institution to ensure that they met the delineation consensus guidelines[7]. Next, three senior physicians (C. Lin, X. Ye, and J. Ge) further examined all cases. If any cases in an institution were found deviating from the delineation guideline, the modification suggestions were provided to corresponding senior physicians of that institution for confirmation and follow-up editing. Detailed patient statistics and subject characteristics of these five external institution datasets are given in Table 1.

**Multi-user testing dataset.** To further evaluate the clinical applicability of SOARS, 50 nasopharyngeal cancer (NPC) patients were randomly selected from two external institutions (30 from FAH-ZU and 20 from SMU) to form a multi-user testing dataset. In this cohort, for each patient, we used 13 common OAR reference contours of FAH-ZU and SMU, the tumor target volume contours, and the IMRT plan originally

generated by the clinical teams. First, two senior physicians edited the SOARS predicted 42 OARs (resulting in SOARS-revised contours) and recorded the editing time to assess the revision efforts required for making SOARS predicted OAR contours to be clinically accepted. One senior physician manually edited the 13 common OARs used in FAH-ZU and SMU, while the other senior physician edited the rest 29 OARs. Second, another physician with 4 years' experience manually contoured the 13 common OARs used in FAH-ZU and SMU following the consensus guideline[7] (denoted as human reader contours). Then, using the clinical reference contours of the 13 OARs as gold-standard references, we compared the contouring accuracy of SOARS, SOARS-revised, and the human reader. Third, we measured the direct dosimetric accuracy ($\text{Diff}^{\text{direct}}_{\text{mean dose}}$ and $\text{Diff}^{\text{direct}}_{\text{max dose}}$ in Eqs. 6 and 7) brought by the OAR's variance. To do this, we kept the original dose grid in the clinical-treated IMRT plan, we replaced the clinical reference OAR contours with SOARS, SOARS-revised, and human reader contours, respectively, to analyze the direct impact on OARs' dose metrics. Fourth, we further examined the clinical dosimetric accuracy ($\text{Diff}^{\text{clinical}}_{\text{mean dose}}$ and $\text{Diff}^{\text{clinical}}_{\text{max dose}}$ in Eqs. 8 and 9), where three new IMRT planning dose grids were generated by using the original tumor target volumes and three substitute OAR contour sets (SOAR, SOARS-revised, and human reader). Then, the clinical reference OAR contours were overlaid on top of each replanned dose grid to evaluate the dose metrics. We randomly selected 10 patients from FAH-ZU's multi-user

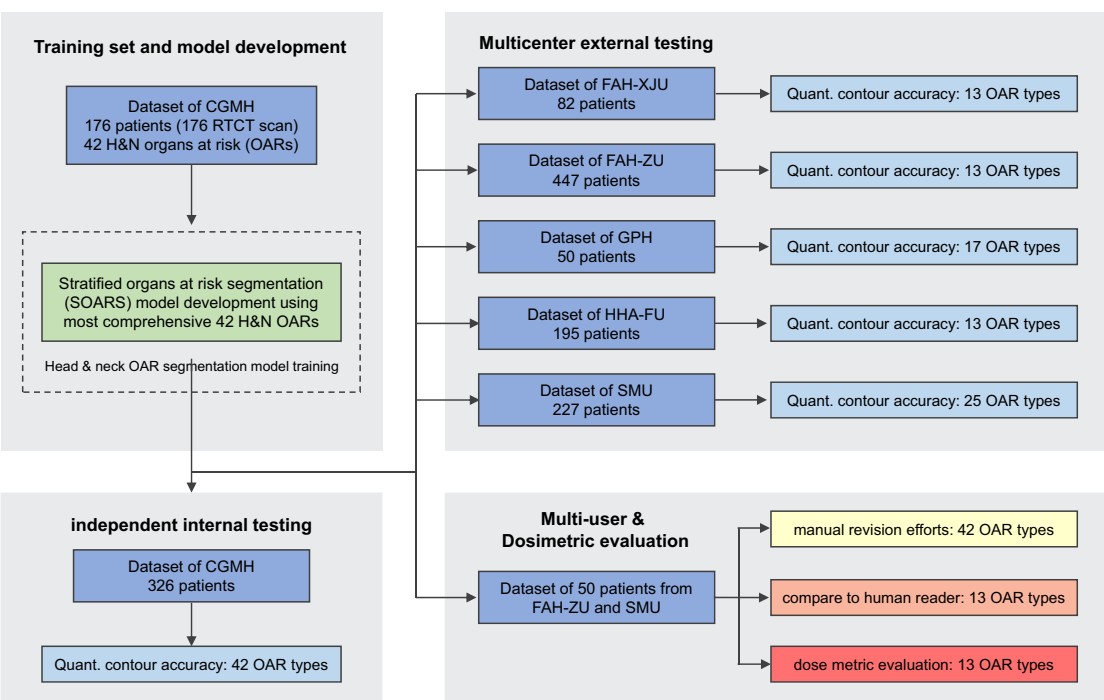

**Fig. 1 | The study flow diagram. We totally collected 1503 head and neck (HN) cancer patients to develop and evaluate the performance of the proposed stratified organ at risk segmentation (SOARS).** The training patients were collected from Chang Gung Memorial Hospital (CGMH), while the testing patients were collected from the internal institution CGMH and other five external institutions including First Affiliated Hospital of Xi'an Jiaotong University (FAH-XJU), First Affiliated Hospital of Zhejiang University (FAH-ZU), Huadong Hospital Affiliated of Fudan University (HHA-FU), Gansu Provincial Hospital (GPH), and Southern Medical University (SMU). We further randomly selected 50 nasopharyngeal cancer patients from FAH-ZU and SMU to form a multi-user testing dataset to evaluate the clinical applicability of SOARS, including the effort for manual revision, comparison to the inter-user OAR segmentation accuracy and comparison to the inter-user OAR dosimetric accuracy.

testing set for this user study. These two dosimetric experiments help determine if differences in OAR contouring would produce clinically relevant differences of radiation doses received by the OARs in the downstream dose planning stage. The overview of the multi-user evaluation is illustrated in Fig. 1.

It is worth noting that clinical reference OAR contours (gold standard OAR contours) of all patients from the independent internal testing dataset, multi-institutional external testing dataset and the multi-user testing dataset do not appear in the training. Training data only includes the training-validation dataset, i.e., 176 patients from CGMH.

**Public HN OAR datasets.** Finally, we evaluated two public HN OAR segmentation datasets to demonstrate the performance of SOARS, i.e., MICCAI[15] and StructSeg 2019 (https://structseg2019.grand-challenge.org) datasets. MICCAI 2015 dataset provides 33 training and 15 testing patients recruited from Norther America, and considers 9 HN OARs: brainstem, mandible (left and right), optic chiasm, optic nerve (left and right), parotid (left and right), SMG (left and right). StructSeg 2019 dataset includes 50 training and 10 testing patients from mainland China, and examines 22 HN OARs: brainstem, eye (left and right), inner ear (left and right), lens (left and right), mandible (left and right), middle ear (left and right), optic chiasm, optic nerve (left and right), parotid (left and right), pituitary, spinal cord, temporal lobe (left and right), TMJ (left and right).

**Performance on the CGMH internal testing dataset**
The quantitative performance of SOARS in the internal testing dataset is summarized in Table 3. SOARS achieved a mean Dice score coefficient (DSC), Hausdorff distance (HD) and average surface distance (ASD) of 74.8%, 7.9 mm, and 1.2 mm, respectively, among 42 OARs. For stratified OAR categories, mean DSC, HD, and ASD for anchor OARs were 86.9%, 5.0 mm and 0.7 mm, respectively; for mid-level OARs were 74.6%, 12.4 mm, and 1.8 mm, respectively; and for S&H were 67.2%, 3.7 mm and 0.7 mm, respectively. In comparison, the previous state-of-the-art H&N OAR segmentation approach UaNet[24] had inferior performance that was statistically significant (DSC: 69.8% vs 74.8%, HD: 8.8 vs 7.9 mm, ASD: 1.6 vs 1.2 mm; all $p < 0.001$). UaNet adopted a modified version of 3D Mask R-CNN[37], which decoupled the whole task into detection followed by segmentation. Although UaNet achieved one of the previous best performances, it lacked dedicated stratified learning to adequately handle a larger number of OARs, possibly accounting for the markedly inferior segmentation accuracy compared to SOARS. Among three stratified OAR categories, S&H OARs exhibited the largest gap between SOARS and UaNet (DSC: 67.2% vs 59.4%, HD: 3.7 vs 4.7 mm, ASD: 0.7 vs 1.2 mm; all $p < 0.001$). This result further confirmed the advantage of SOARS, which employed an adaptively tailored processing workflow and an optimized network architecture towards a particular category of OARs. Figure 2 shows several qualitative comparisons of the internal testing dataset.

**Performance on the multi-institutional external testing dataset**
The overall quantitative external evaluation and the individual external institution evaluation results are shown in Table 4 and Supplementary Tables 2 to 6. SOARS achieved a mean DSC, HD, and ASD of 78.0%, 6.7 mm, and 1.0 mm, respectively, among 25 H&N OARs overall. These represented significant performance improvement ($p < 0.001$) as compared against the UaNet (~4% absolute DSC increase, 16.3% HD reduction, and 28.5% ASD reduction). For individual institutions, average DSC scores of SOARS ranged from 76.9% in FAH-XJU to 80.7% in GPH, while most institutions yielded approximately 78% DSC. HD

**Table 3 | Quantitative comparisons of the internal testing of 326 patients**

| Anchor OARs | UaNet | | | SOARS | | |
|---|---|---|---|---|---|---|
| | DSC | HD (mm) | ASD (mm) | DSC | HD (mm) | ASD (mm) |
| BrainStem | 81.6% ± 5.3% | **8.8 ± 3.3** | 1.7 ± 0.7 | **83.2% ± 5.8%** | 9.0 ± 4.3 | **1.6 ± 0.8** |
| Cerebellum | 90.1% ± 9.4% | 9.5 ± 6.6 | 1.2 ± 0.5 | **92.9% ± 2.2%** | 7.9 ± 4.8 | **0.9 ± 0.3** |
| Eye_Lt | 85.1% ± 13.2% | 3.7 ± 1.4 | 0.8 ± 0.5 | **88.5% ± 4.9%** | 2.9 ± 1.0 | **0.3 ± 0.4** |
| Eye_Rt | 86.2% ± 9.8% | 3.6 ± 1.2 | 0.8 ± 0.4 | **88.7% ± 4.8%** | 2.8 ± 1.1 | **0.3 ± 0.4** |
| Mandible_Lt | 85.0% ± 14.6% | 14.7 ± 13.3 | 1.7 ± 4.4 | **89.1% ± 2.9%** | 5.4 ± 4.4 | **0.5 ± 0.4** |
| Mandible_Rt | 86.0% ± 12.3% | 13.5 ± 11.6 | 1.5 ± 3.9 | **89.0% ± 3.4%** | 5.5 ± 4.6 | **0.5 ± 0.4** |
| SpinalCord | 81.5% ± 9.8% | 17.1 ± 37.5 | 4.2 ± 14.5 | **86.3% ± 4.0%** | 4.7 ± 1.4 | **0.7 ± 0.2** |
| TMJ_Lt | 73.0% ± 7.0% | 4.8 ± 1.6 | 1.2 ± 0.4 | **81.0% ± 9.1%** | 3.5 ± 1.4 | **0.7 ± 0.5** |
| TMJ_Rt | 75.5% ± 7.1% | 4.4 ± 1.6 | 1.1 ± 0.4 | **83.6% ± 6.6%** | 3.4 ± 1.1 | **0.6 ± 0.3** |
| **Mid-level OARs** | | | | | | |
| BasalGanglia_Lt | **76.0% ± 7.7%** | **9.5 ± 3.0** | **1.8 ± 0.7** | 70.9% ± 9.5% | 11.4 ± 3.6 | 2.3 ± 1.0 |
| BasalGanglia_Rt | **73.8% ± 9.1%** | **10.2 ± 2.8** | **2.0 ± 0.8** | 71.4% ± 10.1% | 10.7 ± 3.3 | 2.2 ± 0.9 |
| Brachial_Lt | 57.5% ± 8.1% | **19.8 ± 11.1** | **1.9 ± 1.4** | 60.8% ± 7.6% | 21.8 ± 13.2 | **1.9 ± 1.7** |
| Brachial_Rt | 54.6% ± 10.1% | **19.9 ± 9.7** | **2.0 ± 1.6** | 59.6% ± 7.8% | 24.8 ± 11.7 | 2.0 ± 1.8 |
| Const_Inf | 68.2% ± 11.9% | 7.2 ± 2.9 | 1.3 ± 0.5 | **70.2% ± 10.9%** | 5.9 ± 2.6 | **1.1 ± 0.5** |
| Const_Mid | 61.3% ± 10.8% | 11.5 ± 5.9 | 1.9 ± 0.8 | **63.5% ± 8.7%** | 10.2 ± 5.5 | **1.7 ± 0.6** |
| Const_Sup | 58.6% ± 10.3% | 11.1 ± 4.3 | 2.0 ± 0.9 | **61.2% ± 8.8%** | 10.6 ± 3.9 | **1.9 ± 0.7** |
| Esophagus | **74.2 ± 10.5%** | **16.3 ± 11.4** | **2.0 ± 2.4** | 72.7% ± 11.2% | 28.9 ± 30.6 | 4.2 ± 7.2 |
| Glottic area | 58.6% ± 15.3% | 9.5 ± 6.3 | 2.7 ± 2.1 | **67.8% ± 10.8%** | 6.1 ± 1.9 | **1.7 ± 0.6** |
| GSL | 68.1% ± 10.2% | 8.9 ± 3.6 | 1.3 ± 0.7 | **71.3% ± 9.2%** | 6.7 ± 3.0 | **1.1 ± 0.6** |
| OralCavity | 73.4% ± 6.0% | 21.2 ± 5.1 | 5.1 ± 1.3 | **75.5% ± 7.4%** | 19.2 ± 5.2 | **4.0 ± 1.6** |
| Parotid_Lt | 83.2% ± 5.8% | 9.6 ± 3.3 | 1.4 ± 0.6 | **88.4% ± 4.3%** | 7.8 ± 4.0 | **0.9 ± 0.4** |
| Parotid_Rt | 82.7% ± 6.2% | 10.6 ± 4.6 | 1.5 ± 0.7 | **87.7% ± 3.9%** | 8.4 ± 4.5 | **1.0 ± 0.5** |
| SMG_Lt | 79.2% ± 8.9% | 7.7 ± 4.4 | 1.3 ± 0.6 | **82.0% ± 7.8%** | 6.5 ± 4.1 | **1.0 ± 0.5** |
| SMG_Rt | 77.7% ± 9.2% | 7.9 ± 4.0 | 1.4 ± 0.8 | **82.2% ± 6.6%** | 6.4 ± 2.8 | **1.0 ± 0.4** |
| TempLobe_Lt | 80.9% ± 6.2% | 13.9 ± 5.9 | 2.4 ± 0.9 | **82.9% ± 5.2%** | 13.0 ± 5.0 | **2.2 ± 0.7** |
| TempLobe_Rt | 81.4% ± 5.6% | 13.9 ± 5.1 | 2.3 ± 0.8 | **83.4% ± 5.2%** | 12.1 ± 4.6 | **2.1 ± 0.7** |
| Thyroid_Lt | 80.0% ± 9.8% | **7.5 ± 4.7** | **1.0 ± 0.7** | **82.8% ± 8.7%** | 7.7 ± 15.0 | 1.1 ± 3.5 |
| Thyroid_Rt | 80.6% ± 8.9% | 7.4 ± 4.9 | 1.0 ± 0.9 | **84.1% ± 5.8%** | 6.3 ± 4.0 | **0.8 ± 0.4** |
| **S&H OARs** | | | | | | |
| Cochlea_Lt | 62.8% ± 15.9% | 2.8 ± 1.5 | 0.8 ± 0.7 | **66.0% ± 11.4%** | 2.3 ± 0.7 | **0.6 ± 0.3** |
| Cochlea_Rt | 61.7% ± 16.1% | 2.9 ± 1.6 | 0.8 ± 0.7 | **66.5% ± 10.7%** | 2.3 ± 0.7 | **0.6 ± 0.3** |
| Hypothalamus | 37.5% ± 23.1% | 9.2 ± 4.2 | 3.0 ± 1.9 | **59.1% ± 11.5%** | 5.7 ± 2.2 | **1.4 ± 0.7** |
| InnerEar_Lt | 65.6% ± 11.3% | 4.2 ± 1.6 | 1.1 ± 0.6 | **75.3% ± 7.9%** | 3.0 ± 0.7 | **0.6 ± 0.3** |
| InnerEar_Rt | 66.0% ± 10.4% | 4.2 ± 1.4 | 1.1 ± 0.5 | **75.0% ± 7.8%** | 3.0 ± 0.7 | **0.7 ± 0.6** |
| LacrimalGland_Lt | 45.9% ± 13.7% | 5.7 ± 1.4 | 1.6 ± 0.5 | **57.8% ± 9.5%** | 4.0 ± 0.9 | **0.9 ± 0.3** |
| LacrimalGland_Rt | 43.6% ± 13.9% | 5.6 ± 1.3 | 1.6 ± 0.5 | **56.3% ± 10.2%** | 4.3 ± 1.2 | **1.0 ± 0.3** |
| Lens_Lt | 70.9% ± 8.9% | 2.8 ± 0.7 | 0.6 ± 0.3 | **74.8% ± 9.7%** | 2.7 ± 0.8 | **0.4 ± 0.3** |
| Lens_Rt | 72.4% ± 9.7% | 2.8 ± 0.7 | 0.5 ± 0.3 | **79.5% ± 8.3%** | 2.2 ± 0.8 | **0.3 ± 0.2** |
| OpticChiasm | 59.8% ± 15.8% | 6.5 ± 2.4 | 1.4 ± 0.7 | **67.1% ± 11.4%** | 6.4 ± 2.1 | **0.8 ± 0.5** |
| OpticNerve_Lt | 67.6% ± 8.6% | 5.2 ± 2.6 | 0.8 ± 0.3 | **69.8% ± 7.3%** | 4.8 ± 3.1 | **0.7 ± 0.3** |
| OpticNerve_Rt | 67.0% ± 9.7% | 5.4 ± 4.6 | 0.8 ± 0.5 | **68.2% ± 7.1%** | 4.5 ± 3.0 | **0.7 ± 0.3** |
| PinealGland | 50.6% ± 14.0% | 4.0 ± 1.4 | 1.1 ± 0.5 | **55.6% ± 10.1%** | 3.6 ± 1.3 | **0.9 ± 0.4** |
| Pituitary | 60.2% ± 16.0% | 4.1 ± 1.3 | 1.0 ± 0.4 | **69.6% ± 12.1%** | 3.4 ± 1.2 | **0.6 ± 0.4** |
| Average Anchor | 82.7% | 8.9 | 1.6 | **86.9%** | **5.0** | **0.7** |
| Average Mid-level | 72.1% | 11.8 | 1.9 | **74.6%** | 12.4 | **1.8** |
| Average S&H | 59.4% | 4.7 | 1.2 | **67.2%** | **3.7** | **0.7** |
| **Average all** | 69.8% | 8.8 | 1.6 | **74.8%** | **7.9** | **1.2** |

Note: Bold values represent the statistically significant improvements (calculated using Wilcoxon matched-pairs signed rank test) as compared between UaNet and SOARS. When the mandible is considered as a single OAR instead of left and right mandible, SOARS achieves the mean DSC, HD and ASD of 89.1%, 5.4 mm, and 0.5 mm, respectively. Similarly, when the thyroid is considered as a single OAR instead of left and right thyroid, SOARS achieves the mean DSC, HD, and ASD of 83.5%, 7.0 mm, and 1.0 mm, respectively.

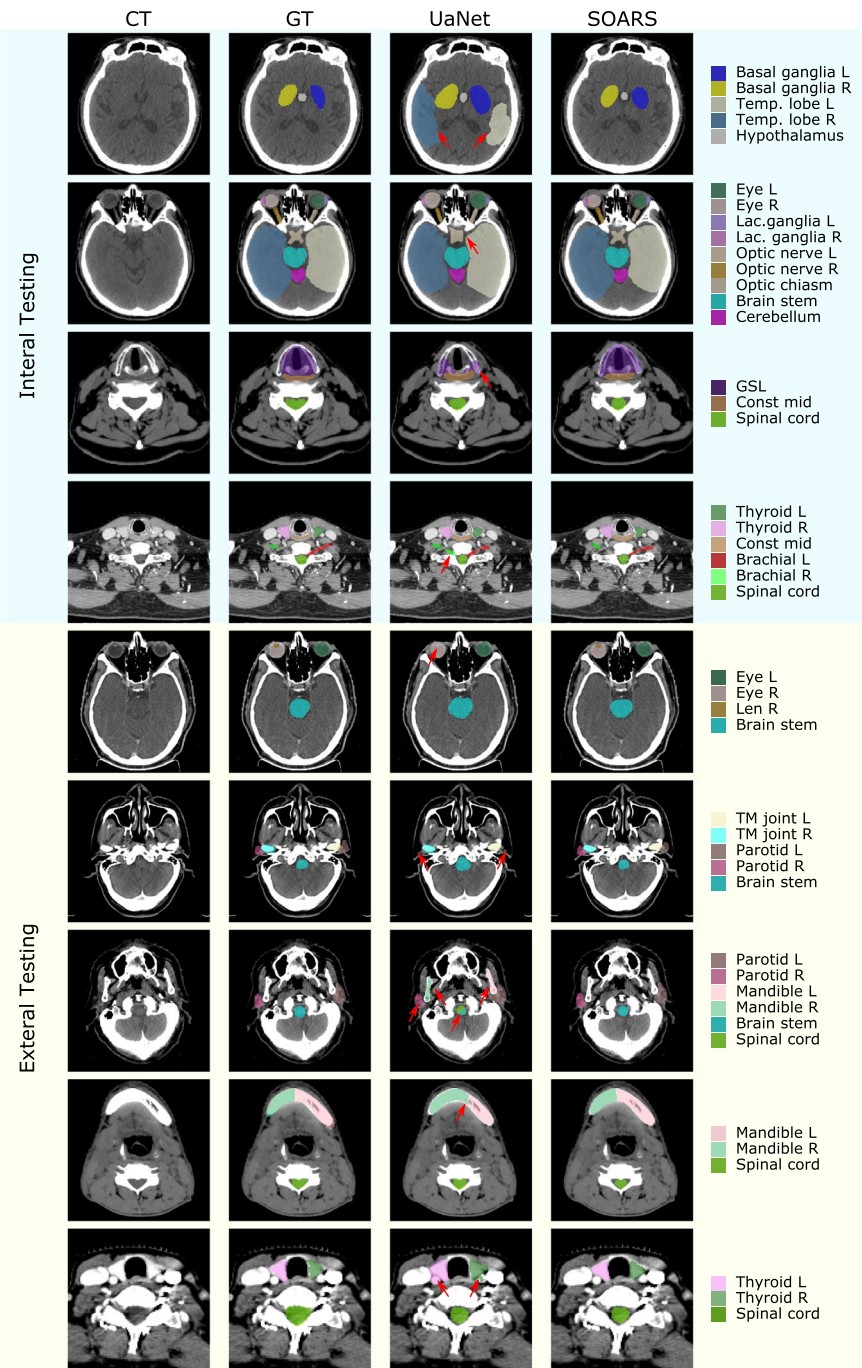

**Fig. 2 | Qualitative 42 OAR segmentation using UaNet and SOARs on the internal (upper 4 rows) & external (lower 5 rows) datasets.** Rows 5–9 are sample images from GPH, FAH-ZU, HHA-FU, SMU, and FAH-XJU, respectively. The 1–4 columns are pCT image, pCT with manual OAR delineations, pCT with UaNet predictions, pCT with SOARS predictions, respectively. The five external centers have different OAR delineation protocols–a subset of 42 OARs is manually labeled. For better comparison, we only show the ground truth associated predictions and use red arrows to indicate the improvements.

values of SOARS were from 5.9 mm in FAH-ZU to 8.1 mm in SMU; and ASD obtained from 0.9 mm in FAH-ZU and GPH to 1.3 mm in SMU and FAH-XJU. Although the OAR numbers varied for external institutions (due to differences among institutional specific RT treatment protocols), these quantitative performance metrics are generally comparable against the internal testing performance levels, demonstrating that SOARS' generality and accuracy hold well to this large-scale external dataset. SOARS consistently and statistically significantly outperforms ($p < 0.001$) UaNet in external evaluation (UaNet had a mean DSC, HD and ASD of 74.3%, 8.0 mm, and 1.4 mm, respectively). SOARS outperforms UaNet in 21 out of 25 OARs on all metrics, with an average DSC improvement of ~4% and relative distance error reductions of 16.3% for HD and 28.5% for ASD.

## Performance on the public datasets

Quantitative evaluation results on the MICCAI 2015 dataset are shown in Supplementary Table 7. When the SOARS model (trained using CGMH Training-Validation dataset and denoted as SOARS_Inference) was directly applied to the MICCAI 2015 testing set, it led to a decent performance of 80.4% mean DSC among 9 OARs higher than most of the recent methods[19–23]. After retraining SOARS using the MICCAI 2015 training set (denoted as SOARS_Retrain), it achieved the top

**Table 4 | Quantitative comparisons on the external testing dataset of 1001 patient**

| OARs | UaNet | | | SOARS | | |
|---|---|---|---|---|---|---|
| | DSC | HD (mm) | ASD (mm) | DSC | HD (mm) | ASD (mm) |
| BrainStem | 77.7% ± 10.7% | 11.4 ± 11.1 | 2.6 ± 2.4 | **81.2% ± 9.8%** | **9.6 ± 11.3** | **2.0 ± 2.3** |
| Eye_Lt | 86.8% ± 5.6% | 3.9 ± 1.4 | 0.7 ± 0.4 | **89.1% ± 4.8%** | **3.7 ± 1.5** | **0.5 ± 0.3** |
| Eye_Rt | 86.6% ± 6.4% | 4.1 ± 4.1 | 0.8 ± 3.1 | **88.9% ± 4.2%** | **3.6 ± 1.0** | **0.5 ± 0.3** |
| InnerEar_Lt | 55.1% ± 12.8% | 8.0 ± 7.4 | 1.9 ± 1.0 | **61.6% ± 14.0%** | **4.9 ± 2.0** | **0.9 ± 0.6** |
| InnerEar_Rt | 54.0% ± 14.5% | 9.4 ± 11.2 | 2.4 ± 2.4 | **64.0% ± 13.8%** | **4.7 ± 1.9** | **0.8 ± 0.5** |
| Lens_Lt | 74.4% ± 11.1% | 2.6 ± 1.0 | 0.5 ± 0.4 | **76.8% ± 9.7%** | **2.5 ± 1.0** | **0.4 ± 0.4** |
| Lens_Rt | 74.7% ± 10.8% | 2.6 ± 1.0 | **0.4 ± 0.5** | **76.9% ± 9.4%** | 2.5 ± 0.9 | 0.4 ± 0.3 |
| Mandible_Lt | 85.5% ± 12.2% | 9.2 ± 8.8 | 1.5 ± 2.7 | **88.9% ± 3.5%** | **7.6 ± 7.5** | **1.2 ± 1.0** |
| Mandible_Rt | 85.8% ± 7.1% | 9.3 ± 8.0 | 1.3 ± 1.2 | **89.2% ± 3.3%** | **7.7 ± 7.6** | **1.2 ± 1.0** |
| OpticChiasm | 55.1% ± 15.6% | 9.1 ± 5.1 | 2.1 ± 1.4 | **66.2% ± 12.3%** | **6.6 ± 4.2** | **1.0 ± 0.6** |
| OpticNerve_Lt | 63.8% ± 12.8% | 7.6 ± 5.2 | 1.1 ± 1.6 | **66.8% ± 8.2%** | **5.3 ± 2.7** | **0.7 ± 0.4** |
| OpticNerve_Rt | 65.5% ± 12.2% | 6.7 ± 4.1 | 1.0 ± 0.9 | **66.6% ± 8.3%** | **5.1 ± 2.3** | **0.7 ± 0.3** |
| OralCavity | 66.4% ± 5.6% | **23.6 ± 3.8** | 5.7 ± 1.0 | **68.5% ± 7.2%** | 25.7 ± 4.5 | **4.8 ± 1.4** |
| Parotid_Lt | 83.2% ± 5.9% | 11.6 ± 6.9 | 1.4 ± 0.8 | **85.7% ± 5.0%** | **10.0 ± 6.9** | **1.1 ± 0.6** |
| Parotid_Rt | 82.8% ± 6.4% | 11.9 ± 8.7 | 1.6 ± 2.1 | **85.2% ± 5.1%** | **10.6 ± 8.2** | **1.2 ± 1.6** |
| Pituitary | 67.5% ± 15.4% | 4.1 ± 1.4 | 0.9 ± 0.7 | **74.7% ± 10.6%** | **3.6 ± 1.1** | **0.5 ± 0.4** |
| SpinalCord | 81.2% ± 10.1% | 10.6 ± 19.4 | 1.3 ± 4.6 | **83.8% ± 7.1%** | **7.2 ± 15.7** | **1.1 ± 4.6** |
| SMG_Lt | 72.0% ± 2.0% | 9.4 ± 4.9 | 2.4 ± 0.3 | **76.8% ± 4.9%** | **6.4 ± 2.3** | **1.3 ± 0.2** |
| SMG_Rt | **75.1% ± 3.2%** | **8.2 ± 4.9** | 1.5 ± 0.4 | 74.8% ± 5.6% | 9.1 ± 4.3 | **0.9 ± 0.1** |
| TempLobe_Lt | 75.9% ± 4.3% | 22.5 ± 6.7 | 2.6 ± 1.1 | **78.7% ± 3.0%** | **20.7 ± 5.8** | **2.2 ± 0.9** |
| TempLobe_Rt | 78.2% ± 4.3% | **20.2 ± 5.8** | **2.1 ± 0.9** | **79.1% ± 3.3%** | 20.4 ± 7.1 | 2.1 ± 0.9 |
| Thyroid_Lt | 73.1% ± 10.0% | 14.8 ± 15.3 | 2.2 ± 2.3 | **74.5% ± 10.4%** | **14.5 ± 16.0** | **2.1 ± 2.6** |
| Thyroid_Rt | 73.6% ± 10.8% | 10.4 ± 5.5 | 1.6 ± 1.2 | **76.4% ± 9.7%** | **9.2 ± 4.6** | **1.4 ± 1.0** |
| TMJ_Lt | 63.7% ± 12.6% | 6.0 ± 4.3 | 1.6 ± 1.1 | **75.3% ± 9.2%** | **4.1 ± 1.5** | **0.7 ± 0.4** |
| TMJ_Rt | 64.9% ± 12.0% | 6.0 ± 4.7 | 1.6 ± 1.2 | **74.0% ± 9.4%** | **4.3 ± 1.9** | **0.8 ± 0.5** |
| FAH-XJU (#82, OAR 13) | 74.8% | 7.2 | 1.2 | **77.3%** | **6.4** | **1.0** |
| FAH-ZU (#447, OAR 13) | 73.7% | 7.5 | 1.3 | **77.4%** | **5.9** | **0.9** |
| GPH (#50, OAR17) | 76.0% | 7.6 | 1.4 | **80.7%** | **6.8** | **0.9** |
| HHA-FU (#195, OAR 13) | 73.5% | 8.0 | 1.5 | **77.7%** | **6.4** | **1.0** |
| SMU (#227, OAR 25) | 73.4% | 9.5 | 1.8 | **76.9%** | **8.1** | **1.3** |
| **Average all** | 74.3% | 8.0 | 1.4 | **78.0%** | **6.7** | **1.0** |

The "#" and "OAR" in each parenthesis denote the number of patients and the number of annotated OARs, respectively. SOARS achieves the best average performance in all metrics among five external centers. DSC, HD, and ASD represent Dice similarity coefficient, Hausdorff distance and average surface distance, respectively.
Note: Bold values represent the statistically significant improvements (calculated using Wilcoxon matched-pairs signed rank test) as compared between UaNet and SOARS, respectively. When the mandible is considered as a single OAR instead of left and right mandible, SOARS achieves the mean DSC, HD, and ASD of 89.1%, 7.6 mm, and 1.2 mm, respectively. Similarly, when the thyroid is considered as a single OAR instead of left and right thyroid, SOARS achieves the mean DSC, HD and ASD of 75.5%, 11.8 mm, and 1.7 mm, respectively.

performance of 83.6% mean DSC with 2.4% absolute DSC improvement over the leading approach of UaNet[24].

Quantitative experimental results on the StructSeg 2019 dataset are shown in the supplementary Table 8. For this dataset, because of the broken-down issue of the official challenge website, it is not feasible to evaluate on the testing set. We chose to conduct a 5-fold cross-validation using the available CT and OAR annotations of 50 patients and compared among SOARS, UaNet[24] and nnUNet[38]. Out of 22 considered OARs, SOARS achieved a mean DSC and HD of 80.9% and 5.8 mm, respectively, outperforming those of UaNet (78.6% and 6.6 mm) and nnUNet (79.2% and 6.7 mm).

**Assessment of editing effort in multi-user testing dataset**
In 50 multi-user evaluation patients, assessment from two senior physicians showed that the vast majority (2060 of 2100 = 42 OAR types × 50, or 98%) of OAR instances produced by SOARS were clinically acceptable or required only very minor revision (no revision: 1228 (58%); revision < 1 minute: 832 (40%)). Only 40 (2%) OAR instances had automated delineation or contouring errors that required 1–3 minutes of moderate modification efforts. None OAR instances required > 3 mi

of major revision. Figure 3 details the assessment results. Another follow-up blinded assessment experiment indicates that these observations are reliable (see the supplementary material). OAR types that needed the most frequent major revisions are hypothalamus, optic chiasm, esophagus, oral cavity, SMG, and temporal lobes. The average total editing time of all 42 OARs for each patient is 10.3 min. Using a random selection of 5 out of 50 patients, two senior physicians also annotated 42 OARs from scratch, which took averaged 106.4 minutes per patient. Thus, the contouring time was significantly reduced by 90% when editing based on SOARS predictions. This observation strongly confirms the added value of SOARS in clinical practice.

**Inter-user contouring accuracy in multi-user testing dataset**
The contouring accuracy of SOARS, SOARS-revised and human reader in the multi-user testing dataset is shown in Table 5. It is observed that SOARS consistently yielded higher or comparable performance in all 13 OARs (commonly used in FAH-ZU's and SMU's RT protocol) as compared to the performance of the human reader (a physician with 4 years' experience). Overall, SOARS achieved statistically significantly improved quantitative results ($p < 0.001$) in mean DSC (80.9% vs

**Table 5 | Quantitative contouring accuracy and direct dosimetric accuracy (Diff$_{mean dose}^{direct}$ and Diff$_{maxdose}^{direct}$) comparison between SOARS, SOARS-revised and the human reader on the multi-user testing dataset of 50 patient**

| Segmentation accuracy | | | | | | |
|---|---|---|---|---|---|---|
| OARs | Human reader | | SOARS | | SOARS-revised | |
| | DSC | HD (mm) | DSC | HD (mm) | DSC | HD (mm) |
| BrainStem | 84.3% ± 5.0% | 7.4 ± 3.6 | **87.0% ± 3.3%** | **5.7 ± 1.6** | **87.9% ± 3.3%** | **5.1 ± 1.6** |
| Eye_Lt | 89.5% ± 3.3% | 3.4 ± 0.7 | **90.0% ± 2.6%** | **3.3 ± 0.7** | 89.9% ± 2.7% | **3.2 ± 0.7** |
| Eye_Rt | 88.2% ± 5.7% | 3.6 ± 1.0 | **89.3% ± 4.3%** | **3.3 ± 0.5** | 89.1% ± 4.4% | 3.3 ± 0.6 |
| Lens_Lt | 74.7% ± 9.1% | 2.5 ± 0.8 | 74.2% ± 7.6% | **2.2 ± 0.6** | **75.9% ± 7.7%** | **2.2 ± 0.6** |
| Lens_Rt | 72.2% ± 9.7% | 2.7 ± 0.8 | 75.4% ± 6.0% | 2.2 ± 0.6 | **76.8% ± 5.7%** | **2.1 ± 0.6** |
| OpticChiasm | 67.6% ± 15.4% | 5.1 ± 1.7 | **74.2% ± 8.9%** | **4.3 ± 1.0** | **77.7% ± 9.5%** | **4.2 ± 1.2** |
| OpticNerve_Lt | 65.3% ± 9.8% | 8.2 ± 3.6 | **73.2% ± 7.4%** | **4.4 ± 1.9** | **74.8% ± 6.8%** | **3.7 ± 1.3** |
| OpticNerve_Rt | 64.9% ± 12.2% | 7.5 ± 4.5 | **72.1% ± 7.1%** | **4.1 ± 1.6** | **73.4% ± 7.9%** | **3.9 ± 1.5** |
| Parotid_Lt | 83.1% ± 4.6% | 12.6 ± 5.3 | **88.2% ± 3.4%** | **7.6 ± 2.8** | **88.3% ± 3.3%** | **7.6 ± 2.8** |
| Parotid_Rt | 82.7% ± 5.2% | 12.3 ± 5.5 | **87.8% ± 3.6%** | **7.5 ± 2.9** | **87.9% ± 3.5%** | **7.2 ± 2.0** |
| SpinalCord | 81.7% ± 6.1% | 9.3 ± 8.1 | **83.7% ± 3.0%** | **4.4 ± 1.3** | **83.9% ± 3.0%** | **3.8 ± 0.7** |
| TMJ_Lt | 74.0% ± 9.9% | 3.5 ± 0.9 | **79.0% ± 8.6%** | **3.2 ± 0.8** | **82.1% ± 9.9%** | **2.8 ± 0.6** |
| TMJ_Rt | 73.6% ± 12.7% | 3.5 ± 1.3 | **77.6% ± 7.3%** | **3.4 ± 0.6** | **81.1% ± 10.5%** | **2.9 ± 0.6** |
| **Average** | 77.1% | 6.3 | **80.9%** | **4.3** | **82.2%** | **4.0** |
| Dosimetric accuracy (Diff$_{mean dose}^{direct}$ and Diff$_{maxdose}^{direct}$) | | | | | | |
| OARs | human reader | | SOARS | | SOARS-revised | |
| | mean dose diff | max dose diff | mean dose diff | max dose diff | mean dose diff | max dose diff |
| BrainStem | 4.7% ± 5.1% | 5.1% ± 6.7% | **2.8% ± 2.8%** | **4.3% ± 4.1%** | 3.1% ± 3.1% | **4.1% ± 4.2%** |
| Eye_Lt | 3.8% ± 3.4% | 6.9% ± 7.5% | 4.1% ± 3.0% | **5.9% ± 4.7%** | 4.4% ± 2.9% | **4.7% ± 4.3%** |
| Eye_Rt | 5.4% ± 5.2% | 6.6% ± 5.7% | 5.1% ± 4.0% | **6.2% ± 5.0%** | 4.9% ± 3.9% | 5.8% ± 5.0% |
| Lens_Lt | 2.2% ± 2.7% | 2.8% ± 2.9% | 1.8% ± 2.2% | **2.7% ± 2.7%** | 1.5% ± 1.4% | **2.4% ± 2.4%** |
| Lens_Rt | 3.5% ± 4.2% | 4.8% ± 5.6% | 2.6% ± 3.3% | **3.8% ± 5.8%** | 2.5% ± 3.2% | **3.5% ± 5.7%** |
| OpticChiasm | 8.2% ± 11.5% | 5.9% ± 9.6% | 5.1% ± 6.3% | **3.3% ± 7.0%** | 4.9% ± 8.0% | **3.8% ± 7.3%** |
| OpticNerve_Lt | 13.0% ± 11.2% | 7.2% ± 9.5% | **9.0% ± 9.4%** | **4.0% ± 6.0%** | 9.2% ± 8.8% | **3.5% ± 7.4%** |
| OpticNerve_Rt | 11.7% ± 10.9% | 6.9% ± 9.4% | **9.8% ± 10.1%** | **3.7% ± 4.9%** | 10.9% ± 10.4% | **3.9% ± 6.1%** |
| Parotid_Lt | 4.8% ± 4.8% | 2.1% ± 2.5% | **2.6% ± 3.3%** | **1.4% ± 1.6%** | 2.5% ± 3.2% | **1.5% ± 1.7%** |
| Parotid_Rt | 4.8% ± 5.3% | 2.0% ± 2.2% | **2.8% ± 3.5%** | **1.2% ± 1.9%** | 2.8% ± 3.5% | **1.2% ± 1.9%** |
| SpinalCord | 10.2% ± 12.7% | 3.3% ± 5.4% | **2.9% ± 4.4%** | **1.7% ± 2.3%** | 2.7% ± 4.2% | **1.9% ± 2.3%** |
| TMJ_Lt | 2.6% ± 2.4% | 2.7% ± 2.9% | 3.0% ± 2.5% | **3.1% ± 2.6%** | 2.6% ± 3.1% | **3.2% ± 3.0%** |
| TMJ_Rt | 2.5% ± 3.0% | 2.5% ± 2.7% | **2.8% ± 1.9%** | **1.9% ± 1.9%** | 2.7% ± 2.9% | **2.0% ± 2.1%** |
| **Average** | 6.0% | 4.4% | **4.2%** | **3.3%** | **4.2%** | **3.2%** |

Note: mean dose diff and max dose diff represent the difference in the mean dose and difference in maximum dose, respectively. Bold values represent the best performance scores. DSC, HD and ASD represent Dice similarity coefficient, Hausdorff distance, and average surface distance, respectively. Direct mean dose differences Diff$_{mean dose}^{direct}$ and direct maximum dose differences Diff$_{maxdose}^{direct}$ are calculated using the Eqs. (6) and (7), respectively. DSC higher the better, while HD, Diff$_{mean dose}^{direct}$ and Diff$_{maxdose}^{direct}$ lower the better

77.1%), HD (4.3 vs 6.3 mm) and ASD (0.7 vs 1.0 mm). 11 out of 13 OAR types demonstrated marked improvements when comparing SOARS with the human reader. On the other hand, by comparing the contouring accuracy between SOARS and SOARS-revised, they have shown very similar quantitative performances (mean DSC: 80.9% vs 82.2%, HD: 4.3 vs 4.0 mm, and ASD: 0.7 vs 0.6 mm). Note that SOARS derived contours (both SOARS and SOARS revised) overall have statistically significantly better performance as compared to those of the human reader, representing the inter-user segmentation variation. Results from the inter-user variation and the previous revision effort assessment validated that SOARS can be readily serving as an alternative "expert" to output high-quality automatically delineated OAR contours, where very minor or no manual efforts are usually required on further editing the SOARS' predictions.

**Direct and clinical dosimetric accuracy in multi-user testing dataset**

Although OAR contouring accuracy reflects the OAR delineation quality, we can further examine its impact on the important downstream dose planning step. Two dosimetic experiments were conducted, i.e., the direct dosimetric accuracy (fixing the original clinical dose grid and replacing with the substitute OAR contours) and the clinical dosimetric accuracy (generating the replanned dose grids with substitute OAR contours). The quantitative direct dosimetric accuracy (Diff$_{mean dose}^{direct}$ and Diff$_{dose}^{direct}$) of various OAR sets, i.e., SOARS, SOARS-revised, and human reader, is illustrated in Table 5 and Fig. 4c, and the relationship between contouring accuracy and direct dosimetric accuracy is plotted in Supplementary Figs. 2 and 3. It was observed that, for SOARS, the direct dosimetric differences in mean dose and in maximum dose were 4.2% and 3.3%, respectively, averaged across all 13 OARs using 50 patients. These were statistically significantly smaller ($p < 0.001$) than those of the human reader contours (6.0% and 4.4%), and comparable to those of SOARS-revised (4.2% and 3.2%). More specifically, using SOARS predictions, only 62 out of 650 (9.5%) OAR instances among 50 patients had a mean dose variation larger than 10%, and only 5 OAR instances have a mean dose difference larger than 30%. In comparison, using the human reader contours, 115 out of 650 (17.7%) OAR instances among 50 patients had a mean dose variation larger than 10%, and 20 OAR instances with a mean dose difference larger

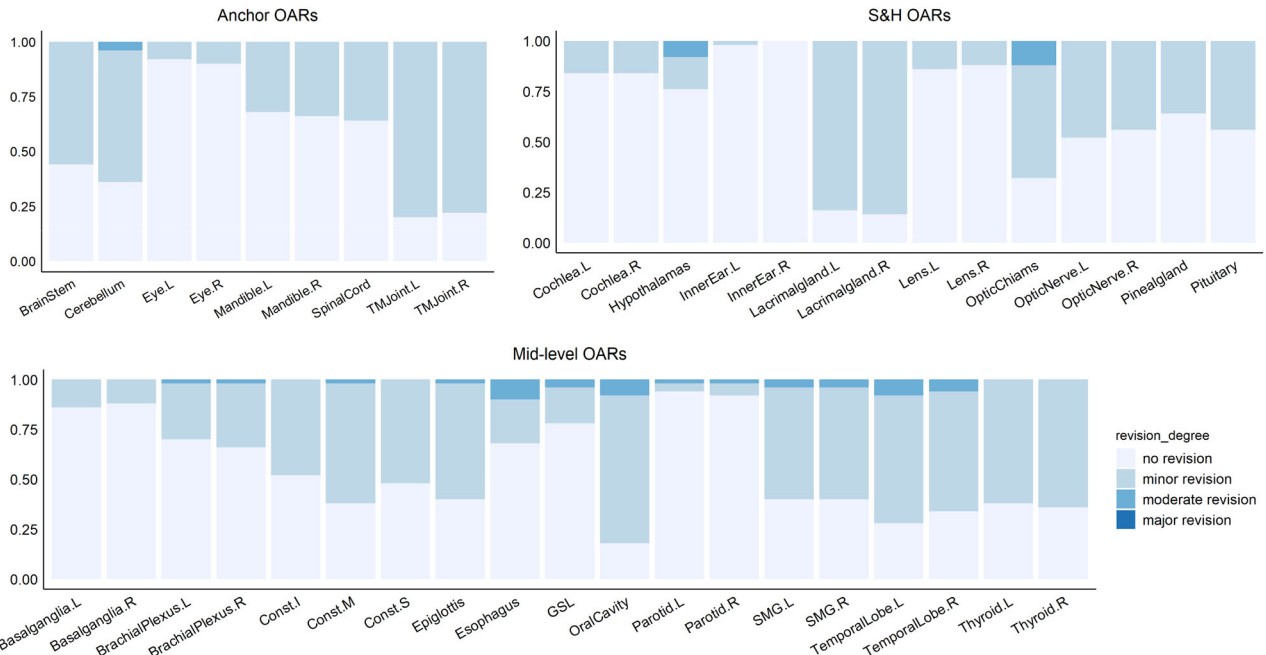

**Fig. 3 | Summary of human experts' assessment of revision effort on SOARS predicted 42 OARs using 50 multi-user testing patients.** Anchor, mid-level and S&H OAR categories are shown separately. Vast majority of SOARS predicted OARs only required minor revision or no revision from expert's editing before they can be clinically accepted. Only a very small amount or percentage of OARs need moderate revision, and no OARs need major revision. Minor revision: editing required in <1 min; moderate revision: editing required in 1–3 min; and major revision: editing required in >3 min.

than 30%. SOARS-revised contours generally had comparable performance with SOARS. Similar trends were observed for the differences in maximum dose (Fig. 4).

For the clinical dosimetric accuracy evaluation ($\text{Diff}^{\text{clinical}}_{\text{mean dose}}$ and $\text{Diff}^{\text{clinical}}_{\text{maxdose}}$), the quantitative results are shown in the supplementary Table 9, and the relationship between the contouring accuracy and clinical dosimetric accuracy is plotted in supplementary Figs. 4 and 5. It was observed that, for SOARS, the clinical dosimetric differences in mean dose and in maximum dose were 5.0% and 3.4%, respectively, averaged across all 13 OARs. These errors were slightly smaller than those of the human reader contours (5.3% and 4.1%, respectively), and comparable to those of SOARS-revised (5.0% and 3.5%, respectively). More OARs from the human reader have clinical dose errors that were larger than 10% or 30%, as compared to SOARS and SOARS-revised, which is consistent with what observed in the direct dosimetric errors.

These results demonstrated that the high contouring accuracy of SOARS led to the better dosimetric accuracy in the dose planning stage. Figure 4(a, b) shows qualitative dosimetric examples and dose-volume histograms (DVH) for using three substitute OAR sets (SOARS, SOARS-revised, human reader). It was observed that doses received by most OARs from SOARS and SOARS-revised matched more closely to the clinical reference doses than those from the human reader.

## Discussion

In this multi-institutional study, a Stratified OAR Segmentation deep learning model, SOARS, was proposed and developed that can be used to automatically delineate 42 H&N OARs following the most comprehensive clinical protocol. By stratifying the organs into three different OAR categories, the processing workflows and segmentation architectures (computed by NAS) were optimally tailored. As such, SOARS is a well-calibrated synthesis of organ stratification, multi-stage segmentation, and NAS. SOARS was trained using 176 patients from CGMH and extensively evaluated on 1327 unseen patients from six institutions (326 from CGMH and 1001 from five other external

medical centers). It achieved a mean DSC and ASD of 74.8% and 1.2 mm, respectively, in 42 OARs from the CGMH internal testing and generalized well to the external testing with a mean DSC of 78.0% and ASD of 1.0mm, respectively, in 25 OARs. SOARS consistently outperformed the previous state-of-the-art method UaNet[24] by 3–5% absolute DSC and 17–36% of relative ASD in all six institutions. In a multi-user study, 98% of SOARS-predicted OARs required no revision or very minor revision from physicians before they were clinically accepted, and the manual contouring time can be reduced by 90% (from 106.4 to 10.3 minutes). In addition, the segmentation and dosimetric accuracy of SOARS were comparable to or smaller than the inter-user variation. It is also noted that the proposed SOARS may be also applied to other body sites, where anatomical structures are densely distributed with different levels of segmentation difficulty, e.g., many anatomical tissues in chest or abdomen regions[39].

Recent consensus guidelines recommend delineating more than 40 OARs in H&N cancer patients[7]. However, in practice, it is an unmet need. Most institutions only delineated a small subset of H&N OARs per their institutional specific RT protocol, or they can only afford to delineate OARs that are closest to tumor targets. The challenges of following the consensus guidelines were probably due to the lack of efficient and accurate OAR delineation tools (most automated tools focused on segmenting less than or around 20 H&N OARs[18,21,23,40]). Manually contouring 40+ OARs was too time-consuming and expertise-demanding, hence unrealistic in practice. Without assessment of the dosimetric results in the complete set of OARs, it was infeasible to track and analyze the organ-specific adverse effects after RT treatment in multi-institutional clinical trials. In addition, data pooling analysis of radiation therapy from different institutions was impeded by the inconsistency in OAR contouring guidance. The Global Quality Assurance of Radiation Therapy Clinical Trials Harmonization Group (CHG) has provided standardized nomenclature for clinical trial use to address this problem[41]. With the proposed SOARS, it is

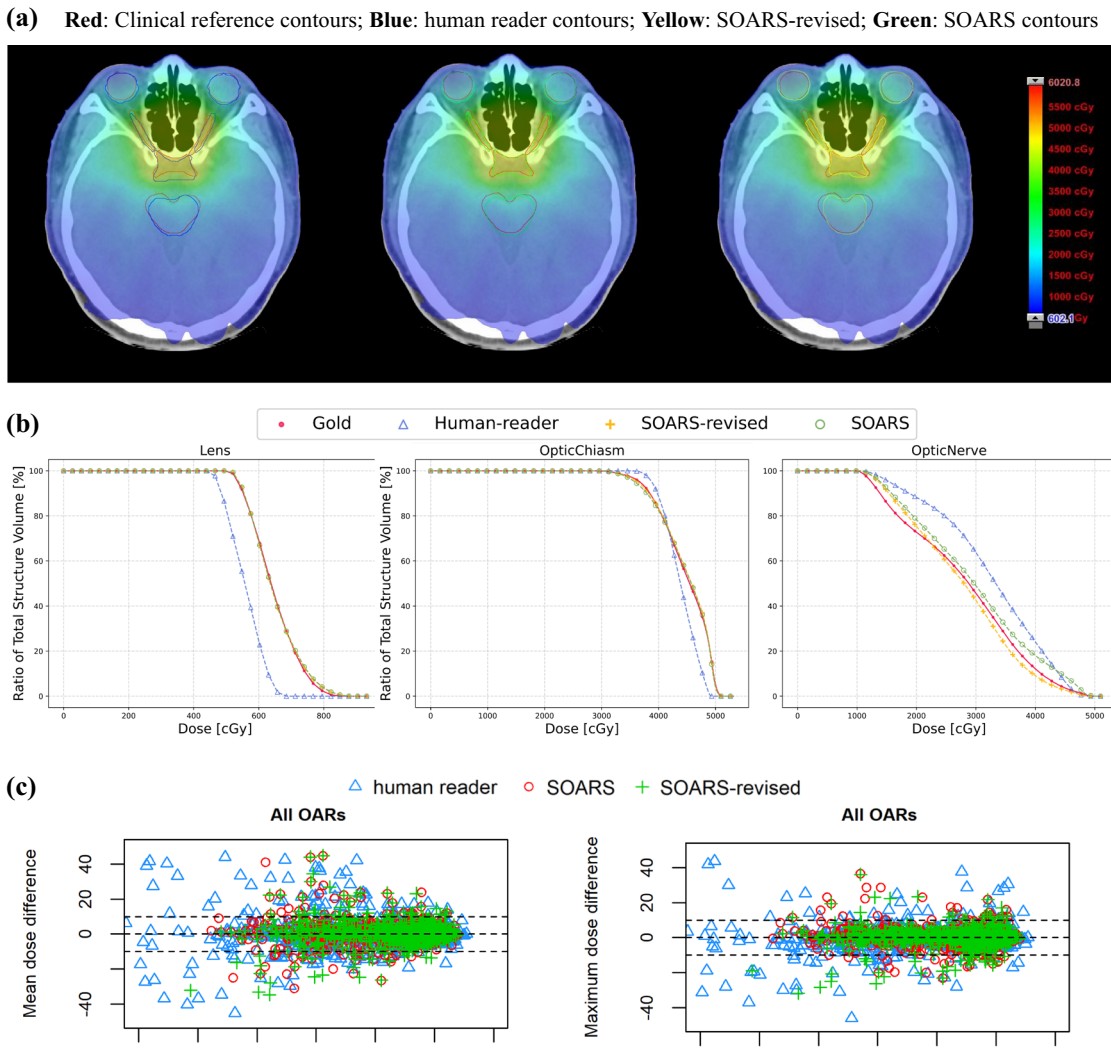

**(a)**  **Red**: Clinical reference contours; **Blue**: human reader contours; **Yellow**: SOARS-revised; **Green**: SOARS contours

**Fig. 4 | OAR dosimetric illustrations.** Using a specific patient, without loss of generality, we illustrate a qualitatively direct dosimetric example **a** in axial views of two anatomic locations. Clinical OAR reference: red; human reader: blue; SOARS OAR: green; SOARS-revised: yellow. **b** the dose–volume histograms (DVH) plot of OARs in this patient. **c** The scatter plots of direct mean dose differences $\mathrm{Diff}^{\mathrm{direct}}_{\mathrm{mean\,dose}}$ and direct maximum dose differences ($\mathrm{Diff}^{\mathrm{direct}}_{\mathrm{maxdose}}$) brought by various

OAR contour sets of human reader, SOARS, and SOARS-revised among 50 multi-user testing patients. Blue triangle, green cross and red circle represent the results of human reader, SOARS-revised and SOARS, respectively. Wilcoxon matched-pairs signed rank test is used to compare between SOARS/SOARS-revised results and human reader's results, and statistical significance is set at two-tailed $p < 0.05$.

feasible to provide comprehensive OAR dose evaluation, further facilitating post-treatment complications and quality assurance studies.

In this work, from the OAR contouring quality, we further analyzed the OAR dosimetric accuracy in the subsequent dose planning step. Two dosimetric evaluation scenarios were considered and analyzed, i.e., (1) the direct dosimetric evaluation when fixing the original clinical reference dose grid and replacing the clinical reference OAR contours with those of substitute OARs; and (2) the clinical dosimetric evaluation when generating new replanned dose grids with substitute OAR contours and overlaying the clinical reference OAR contours on top of each. The dosimetric differences in mean dose and in maximum dose for both scenarios were used as dose metrics consistently with previous work[42]. Overall, the majority of SOARS-predicted OAR instances had the mean and maximum dose variance no larger than 10%, which was comparable to or smaller than the inter-user dose variations in our experiment. This variation was also smaller than the previously reported inter-user dose variations in six H&N OAR types[42],

where quite few are larger than 30% or even above 50%. For individual OARs, we observed that the optic chiasm and optic nerve (left and right) exhibited increased dose variation (10–40%) in a small portion of patients (Supplementary Figs. 2 to 5). This phenomenon was consistently observed in SOARS, SOARS-revised, and the human reader contours. This indicated that dosimetries in areas consisting of these OARs are sensitive to contouring differences, suggesting that more attention should be required to delineate the above OAR types for NPC patients.

It is also worth noting that in the generation of clinical reference OAR contours ("gold standard" contours for training and validation) of our study, MRI scans (if available) and other clinical information were also provided to physicians as reference. We did not directly fuse the pCT and MRI scans in our study. This is because hyperextension positioning under the cast fixation for CT simulation is usually used in head and neck cancer treatment, while diagnostic MR images are acquired in a neutral position. Directly fusing them using the current rigid or deformable registration algorithms often leads to large

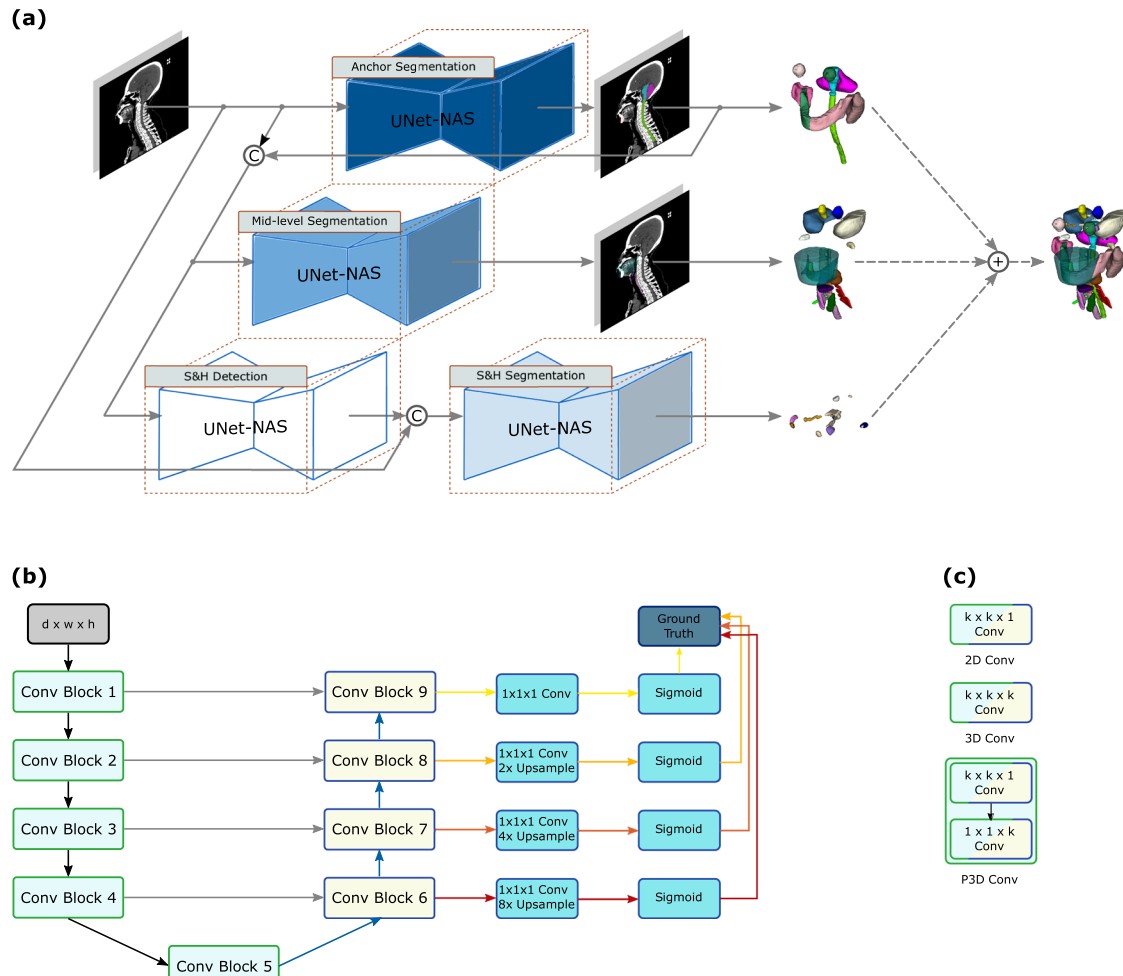

**Fig. 5 | The stratified organ at risk segmentation (SOARS) method. a** SOARS first stratifies OAR into anchor, mid-level, and small & hard (S&H) categories and uses the anchor OARs to guide the mid-level and S&H OAR segmentation. **b** SOARS further stratifies the backbone network UNet with neural architecture search (NAS), which permits an automatic selection across 2D, 3D, and P3D convolution blocks. **c** The NAS convolution setting.

errors[43]. Instead, physicians would open two PACS windows in the computer to view pCT and MRI separately to help the delineation (if they felt it necessary to consult to MRI). Hence, the "gold standard" OAR references could be viewed as human experts contouring using both CT and MR information. In contrast, SOARS is trained using only CT images (with the "gold standard" labels involving both CT and MRI) and is able to reliably generate the OAR contours on unseen patients using only CT. This may be the strength of deep learning methods, which could use CT modality alone to achieve statistically comparable or in some scenarios better and/or more consistent performance than human experts leveraging on both CT and MRI.

Our study has several limitations. First, the external testing datasets do not have a complete set or the same amount of 42 OAR types. This reflects real-world situations among different institutions. Manually labeling 42 OARs for all 1001 external testing patients is impractical (estimated to require ≥3 h per patient). Hence, we chose to use the existing clinically labeled OAR types to supplement testing. Second, the multi-user testing dataset of FAH-ZU contains only 13 clinical reference OAR types according to its RT protocol. Thus we evaluated the inter-user variation of segmentation and dosimetric accuracy using these 13 OARs instead of the complete 42 OAR types. Nevertheless, these 13 OAR types included those from the three different OAR categories of anchor, mid-level, and S&H. We believe the performance from these would reflect the real inter-user variation with a larger number of OAR types.

Third, it would be helpful to conduct a randomized clinical trial comparing the side effects and life quality as outcomes of manual and SOARS-assisted OAR contouring. This could further validate the clinical value of SOARS. We leave this for our future works.

To conclude, we introduced and developed a stratified deep learning method to segment the most comprehensive 42 H&N OAR types in radiotherapy planning. Through extensive multi-institutional validation, we demonstrated that our SOARS model achieved accurate and robust performance and produced comparable or higher accuracy in OAR segmentation and the subsequent dose planning than the inter-user variation. Physicians needed very minor or no revision for 98% of the OAR instances (when editing on SOARS predicted contours) to warrant clinical acceptance. SOARS could be implemented and adopted in the clinical radiotherapy workflow for a more standardized, quantitatively accurate, and efficient OAR contouring process with high reproducibility.

## Methods

Requirements to obtain informed consent were waived by the institutional review boards because this study is retrospective and does not affect patients' treatment and outcomes. A total of 1503 patients with head and neck cancer from six institutions were collected in this retrospective study under each institutional review board approval, including Chang Gung Memorial Hospital, First Affiliated Hospital of

Xi'an Jiaotong University, First Affiliated Hospital of Zhejiang University, Gansu Provincial Hospital, Huadong Hospital Affiliated of Fudan University, and Southern Medical University.

The SOARS framework is illustrated in Fig. 5. It consists of three processing branches to stratify the anchor, mid-level, and S&H OAR segmentation, respectively. Stratification manifested first in the distinct processing workflow used for each OAR category. We next stratified neural network architectures by using differentiable neural architecture search (NAS)[26,27] to search a distinct network structure for each OAR category. We will explain each stratification process below.

## Processing stratification in SOARS

SOARS first segmented the anchor OARs. Then, with the help of predicted anchor OARs, mid-level and S&H OARs were segmented. For the most difficult category of S&H OARs, SOARS first detected their center locations and then zoomed in accordingly to segment the small OARs. For the backbone of all three branches, we adopted the UNet structure implemented in the nnUNet framework[38], which has demonstrated leading performance in many medical image segmentation tasks. We tailored each UNet with NAS, which is explained in the subsequent subsection.

We denoted the training data of $N$ instances as $S = \left\{ X_i, Y_i^A, Y_i^M, Y_i^S \right\}_{i=1}^{N}$, where $X_i$, $Y_i^A$, and $Y_i^S$ were the input pCTs and ground-truth masks for anchor, mid-level, and S&H OARs, respectively. The indexing parameter $i$ was dropped for clarity. We used boldface to denote vector-valued volumes and used vector concatenation as an operation across all voxel locations.

*Anchor branch:* Assuming there are $C$ anchor classes, we first used the anchor branch to generate OAR prediction maps for every voxel location, $j$, and every output class, $c$:

$$\hat{Y}_c^A(j) = p^A(Y^A(j) = c | X; W^A), \hat{Y}^A = [\hat{Y}_1^A \cdots \hat{Y}_C^A] \qquad (1)$$

where UNet functions, parameters, and the output prediction maps were denoted as $p^A(\cdot)$, $W(\cdot)$ and $\hat{Y}^A$, respectively. Anchor OARs are easy and robust to segment based on their own CT image appearance and spatial context features. Consequently, they provided highly informative location and semantic cues to support the segmentation of other OARs.

*Mid-level branch:* Most mid-level OARs are primarily soft tissue, which has limited contrast and can be easily confused with other structures with similar intensities and shapes. Hence, we incorporated the anchor predictions into mid-level learning. Specifically, the anchor predictions and the pCT were concatenated to create a multi-channel input $[X, \hat{Y}^A]$:

$$\hat{Y}_c^M(j) = p^M(Y^M(j) = c | X, \hat{Y}^A; W^M) \qquad (2)$$

*Small & hard branch:* Considering the low contrast and unbalanced class distributions for S&H OARs across the entire CT volume, direct S&H OAR segmentation is challenging. Here, we further decoupled this branch into a detection followed by segmentation process. Because the H&N region has relatively stable anatomical spatial distribution, detecting rough locations of S&H OARs is a much easier and reliable task. Once the OAR center was approximately determined, a localized region can be cropped out to focus on segmenting the fine boundaries in a zoom-in fashion. The detection was implemented using a simple yet effective heat map regression approach and the heat map labels were generated at each organ center using a 3D Gaussian kernel[44,45]. Let $f(\cdot)$ denote the UNet function for the detection module, we also combined the anchor branch predictions with pCT as the detection input:

$$\hat{H} = f(X, \hat{Y}^A; W^D), \qquad (3)$$

where $\hat{H}$ were the predicted heat maps of S&H OARs. Given the regressed heat map $\hat{H}$, the pixel location corresponding to the highest value was extracted to crop a volume of interest (VOI) using three times the extent of the maximum size of the OAR of interest. Then, SOARS segmented the fine boundaries of S&H OARs within the VOI. Let $V$ denote the cropped VOI in pCT. The S&H OAR segmentation was implemented as:

$$\hat{Y}_c^S(j) = p^S(Y^S(j) = c | V; W^S). \qquad (4)$$

## Automatic neural architecture search in SOARS

Considering the significant statistical variations in OAR appearance, shape, and size, it is unlikely that the same network architecture would suit each OAR category equally. Hence, SOARS automatically searches the more suitable network architectures for each branch, adding an additional dimension to the stratification. We conducted the differentiable NAS[26,27] on top of the network structure of UNet[29]. The NAS search space included 2D, 3D, and pseudo-3D convolutions with either kernel sizes of 3 or 5. Figure 5b, c demonstrates the network architecture and the search space of NAS. Let $\phi(\cdot; \omega_{x \times y \times z})$ denote a composite function of the following consecutive operations: a convolution with an $x \times y \times z$ dimension kernel, an instance normalization, and a Leaky ReLu unit. If one of the kernel dimensions is set to 1, it reduces to a 2D kernel. The search space $\Phi$ can be represented as.

$$\phi_{2D_3} = \phi(\cdot; \omega_{3 \times 3 \times 1}),$$

$$\phi_{2D_5} = \phi(\cdot; \omega_{5 \times 5 \times 1}),$$

$$\phi_{3D_3} = \phi(\cdot; \omega_{3 \times 3 \times 3}),$$

$$\phi_{3D_5} = \phi(\cdot; \omega_{5 \times 5 \times 5}),$$

$$\phi_{P3D_3} = \phi(\phi(\cdot; \omega_{3 \times 3 \times 1}); \omega_{1 \times 1 \times 3}),$$

$$\phi_{P3D_5} = \phi(\phi(\cdot; \omega_{5 \times 5 \times 1}); \omega_{1 \times 1 \times 5}),$$

$$\Phi = \{\phi_{2D_3}, \phi_{2D_5}, \phi_{3D_3}, \phi_{3D_5}, \phi_{P3D_3}, \phi_{P3D_5}\} \qquad (5)$$

The architecture was learned in a differentiable fashion. We made the search space continuous by relaxing the selection of $\phi(\cdot; \omega_{x \times y \times z})$ to a softmax function over $\phi$. For $k$ operations, we define a set of $\alpha_k$ learnable logits for each. The weight $\gamma_k$ for an operation is defined as $\gamma_k = \frac{\exp(\alpha_k)}{\sum_m \exp(\alpha_m)}$, and the combined output is $\phi' = \sum_k \gamma_k \phi_k$. As the result of NAS, we selected the operation with the top weight to be the searched operation. We used the same scheme to search the segmentation network architecture for all three branches (excluding the S&H detection module) and trained SOARS using the final auto-searched architecture. The searched network architectures for each branch are listed in supplementary Fig. 1. The implementation details are also reported in the supplementary materials.

## Quantitative evaluation of contouring accuracy

For the internal and external testing datasets, the contouring accuracy was quantitatively evaluated using three common segmentation metrics[46,47], i.e., Dice similarity coefficient (DSC), Hausdorff distance (HD) and average surface distance (ASD). Additionally, for quantitative comparison, we also trained and tested the previous state-of-the-art

H&N OAR segmentation method, UaNet[24]. For the model development of UaNet, we used the default parameter setting from original authors[24] as these have been already specifically tuned for the head and neck OARs. We applied the same training-validation split as ours to ensure a fair comparison.

## Human experts' assessment of revision efforts

An assessment experiment by human experts was conducted to evaluate the editing efforts needed for the predicted OARs to be clinically accepted. Specifically, using the 50 multi-user testing dataset, two senior physicians (X. Ye and J. Ge) were asked to edit SOARS predictions of 42 OARs according to the consensus guideline[7]. Besides the pCT scans, other clinical information, and imaging modality such as MRI (if available) were also provided to physicians as reference. The edited OAR contours were denoted as SOARS-revised. Four manual revision categories were designated as no revision required, revision required in <1 minute (minor revision), revision required in 1–3 minutes (moderate revision), and revision required in >3 minutes (major revision).

## Inter-user contouring evaluation

Using the multi-user testing dataset, we further asked a board-certified radiation oncologist with 4 years' experience specialized in treating H&N cancers to manually delineate the 13 common OAR types used in FAH-ZU and SMU following the consensus delineation guideline[7]. Patients' pCT scans along with their clinical information and other available medical images (including MRI) were provided to the physician. The labeled OAR contours were denoted as human reader contours. Then, we compared the contouring accuracy between SOARS, SOARS-revised, and the human reader using the evaluation metrics of DSC, HD, and ASD. The contouring performance of SOARS-revised and the human reader represents the inter-user variation in OAR contouring.

## Inter-user direct and clinical dosimetric evaluation

Differences in the OAR contouring accuracy would not, by itself, indicate whether such differences are clinically relevant in terms of radiation doses received by the OARs. Therefore, we further quantified the dosimetric impact brought by the OAR contouring differences. Two dosimetric experiments were conducted: (1) the direct dosimetric evaluation by fixing the original clinical dose grid and replacing the clinical reference OAR contours with substitute OAR contours of SOARS, SOARS-revised and human reader; (2) the clinical dosimetric evaluation by generating the replanned dose grids with substitute OAR contours and then overlaying the clinical reference OAR contours on top of each replanned dose grid. Regarding the direct dosimetric evaluation, for each patient in the multi-user testing dataset, we first used the original clinical reference OARs and the corresponding dose grid (dose voxel sizes ranging from 2 to 4 mm) to compute the OAR dose metrics in terms of mean doses and max doses. Then, the same dose grid was combined with different OAR contour sets, i.e., SOARS, SOARS-revised, human reader, and the dose metrics of each OAR contour set were calculated. This design was to isolate the dose effects due strictly to OAR contouring differences because the dose grid was fixed, and the dose metrics were quantified by replacing each clinical reference contours with the substitute contours. Following the work[42], we calculated the direct mean dose and maximum dose differences as follows:

$$\text{Diff}^{\text{direct}}_{\text{mean dose}} = \frac{\text{mean dose}(\text{OAR}_{\text{substitute}}, \text{Dose}_{\text{ref}}) - \text{mean dose}(\text{OAR}_{\text{ref}}, \text{Dose}_{\text{ref}})}{\text{mean dose}(\text{OAR}_{\text{ref}}, \text{Dose}_{\text{ref}})} \times 100\% \quad (6)$$

$$\text{Diff}^{\text{direct}}_{\text{max dose}} = \frac{\text{max dose}(\text{OAR}_{\text{substitute}}, \text{Dose}_{\text{ref}}) - \text{max dose}(\text{OAR}_{\text{ref}}, \text{Dose}_{\text{ref}})}{\text{max dose}(\text{OAR}_{\text{ref}}, \text{Dose}_{\text{ref}})} \times 100\% \quad (7)$$

where $\text{OAR}_{\text{substitute}}$ represents the OAR contours by SOARS, SOARS-revised, and the human reader, respectively, while $\text{OAR}_{\text{ref}}$ and $\text{Dose}_{\text{ref}}$ represent the original clinical reference OAR contours and dose grid used in the original RT plan, respectively. For the clinical dosimetric evaluation, three new IMRT planning dose grids were generated by using the original tumor target volumes and three substitute OAR contours (SOAR, SOARS-revised, and human reader). Then, the clinical reference OAR contours were overlaid on top of each replanned dose grid to calculate the clinical mean dose and maximum dose differences as follows[42]:

$$\text{Diff}^{\text{clinical}}_{\text{mean dose}} = \frac{\text{mean dose}(\text{OAR}_{\text{substitute}}, \text{Dose}_{\text{substitute}}) - \text{mean dose}(\text{OAR}_{\text{ref}}, \text{Dose}_{\text{substitute}})}{\text{mean dose}(\text{OAR}_{\text{ref}}, \text{Dose}_{\text{substitute}})} \times 100\% \quad (8)$$

$$\text{Diff}^{\text{clinical}}_{\text{max dose}} = \frac{\text{max dose}(\text{OAR}_{\text{substitute}}, \text{Dose}_{\text{substitute}}) - \text{max dose}(\text{OAR}_{\text{ref}}, \text{Dose}_{\text{substitute}})}{\text{max dose}(\text{OAR}_{\text{ref}}, \text{Dose}_{\text{substitute}})} \times 100\% \quad (9)$$

where $\text{Dose}_{\text{substitute}}$ represents the new dose grids in the replanned RT when using OAR contours of SOARS, SOARS-revised, and the human reader, respectively. The dose-volume histogram (DVH) was also plotted for qualitative illustration. The dose/DVH statistics were generated using Eclipse 11.0 (Varian Medical Systems Inc., Palo Alto, CA).

## Statistical Analysis

The Wilcoxon matched-pairs signed rank test was used to compare the evaluation metrics in paired data, while Manning-Whitney U test was used to compare the unpaired data. All analyses were performed by using R[48]. Statistical significance was set at two-tailed $p < 0.05$.

## Reporting summary

Further information on research design is available in the Nature Research Reporting Summary linked to this article.

# Data availability

The imaging data from internal and external institutions are not publicly available due to the data privacy and restricted permissions of the current study. The anonymized data are available under restricted access for patient privacy. Access can be obtained by sending a request to the corresponding author for academic purposes. The raw patient data are protected and are not available due to data privacy laws. Sample testing imaging data from two public HN OAR datasets can be directly downloaded from https://www.imagenglab.com/newsite/pddca and https://structseg2019.grand-challenge.org.

# Code availability

The baseline UNet used in this study is implemented in the nnUNet deep learning framework, available at https://github.com/MIC-DKFZ/nnUNet. The codes used for inference and result evaluation is available at: https://doi.org/10.5281/zenodo.6998392.

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

## Acknowledgements

This work is partially supported by Maintenance Project of the Center for Artificial Intelligence in Medicine (Grant CLRPG3H0012, CMRPG3K1091, SMRPG3IO011) at Chang Gung Memorial Hospital.

## Author contributions

For the three first co-authors, X.Y. helped collect the external data, participate, and coordinate the human assessment and contouring analysis in the multi-user studies, D.G. was responsible for the data cleaning, deep learning model development, and the internal and external evaluation, and J.G. helped collect the external data, participate in the human assessment and dosimetric analysis in the multi-user studies, and they all involved in the experimental design and drafted the manuscript. S.Y., Y.Y., Y.S., Z.L., and W.L. participated in the multi-user studies. Y.X. and Z.Z. aided in the deep learning model development and results interpretation. B.H and T.-M.H approved the contours for validation of the internal institution. L.P., Y.R., R.L., G.Z., M.M., and X.C. helped collect, organize, and validated the external data. Y.C. collected and organized internal data. L.H., and J.X. contributed to the design and implementation of the research. A.P.H aided in interpreting the results and edited the manuscript. L. L. aided in the experimental design and interpretation of the results and edited the manuscript. C.L. was responsible for reviewing and modifying the contours from internal institutions, and she also provided guidance and consulting in the multi-user study. D.J. was responsible for the data cleaning, development of the deep learning model, and overseeing the evaluation process. T-Y.H. collected the internal training and evaluation data. D.J. and T.-Y.H. were responsible for the conception and design of the experiments and oversaw overall direction and planning and drafted the manuscript.

## Competing interests

The authors declare no competing interests.

## Additional information

[1]Department of Radiation Oncology, The First Affiliated Hospital, Zhejiang University, Hangzhou, China. [2]DAMO Academy, Alibaba Group, New York, NY, USA. [3]Ping An Technology, Shenzhen, China. [4]Department of Radiation Oncology, Chang Gung Memorial Hospital, Linkou, Taiwan, ROC. [5]Department of Computer Science, Johns Hopkins University, Baltimore, MD, USA. [6]Department of Respiratory Disease, Zhejiang Provincial People's Hospital, Hangzhou, Zhejiang, China. [7]Department of Radiation Oncology, Huadong Hospital Affiliated to Fudan University, Shanghai, China. [8]Department of Radiation Oncology, The First Affiliated Hospital, Xi'an Jiaotong University, Xi'an, China. [9]Department of Radiation Oncology, People's Hospital of Shanxi Province, Shanxi, China. [10]Department of Radiation Oncology, Nanfang Hospital, Southern Medical University, Guangzhou, China. [11]Department of Radiation Oncology, The First Hospital of Lanzhou University, Lanzhou, Gansu, China. [12]Q Bio Inc, San Carlos, CA, USA. [13]Particle Physics and Beam Delivery Core Laboratory, Chang Gung Memorial Hospital and Chang Gung University, Taoyuan, Taiwan, ROC. [14]Department of Nuclear Medicine, Chang Gung Memorial Hospital, Linkou, Taiwan, ROC. [15]These authors contributed equally: Xianghua Ye, Dazhou Guo, Jia Ge. ✉e-mail: qqvirus@cgmh.org.tw; dakai.jin@gmail.com; tyho@cgmh.org.tw

