## [Peer Review File · Nature Communications]

Reviewers' Comments:

Reviewer #1:

Remarks to the Author:

The paper 'Comprehensive and Clinically Accurate Head and Neck Organs at Risk Delineation via Stratified Deep Learning: A Large-scale Multi-Institutional Study' describes the application of a deep-learning based segmentation pipeline to a large cohort of head and neck cancer patients. Segmentation tasks are the 42 OARs that are recommended by international guidelines. The proposed method stratifies these OARs into different subcategories based on size and difficulty and uses a progressive segmentation scheme where predictions of previous steps are incorporated into subsequent ones. Notable is the localization and cropping of surrounding regions of difficult structures to aid the model perform the segmentation accurately. In addition to the stratification, the authors propose to use differentiable neural architecture search (NAS) to find optimal architectures for each of the subcategories.

The proposed model is evaluated extensively and thoroughly: while the training set originates from only a single-institutional cohort, the authors demonstrate that their model generalizes well to patients from multiple external testing datasets. Moreover, the authors carefully evaluate the segmentations with respect to clinical impact in the form of needed revisions and dosimetry. A comparison with the inter-rater variability underlines the strong results.

Overall the paper is very well written and clearly structured. Most of the needed information for reproducing the algorithm can be found in the Online Methods as well as the Supplement.

That said, there are several concerns I would like to raise and invite the authors to comment on:

1) The overwhelming majority of the methodological contributions seems to already have been published as a CVPR paper. The difference to the method presented here is mainly the backbone which was swapped in favor of a U-Net. The other key features of the approach remain the same (stratification, NAS). Thus, the methodological contribution offered by this paper seems limited and the value of the paper mainly lies in its extensive clinically-oriented evaluation. I think the messaging about the key contributions may need rewording.

2) Proposing new methods using only internal datasets is in my opinion no longer sufficient. The paper would greatly benefit from using publicly available datasets for evaluation, for example the MICCAI 2015 challenge dataset (which was also used by the authors in their CVPR publication) and the StructSeg2019 dataset. Unfortunately the website of the latter seems broken at the moment but it should be possible to contact the organizers nonetheless. Independent evaluation on publicly available datasets allows readers (and reviewers) to directly compare the proposed algorithm with the existing literature and thus get a much clearer picture of their capabilities.

3) There seem to be biases in the annotations that favor SOARS over the 'human reader' in the inter-rater disagreement studies. Concretely, both the internal training dataset as well as the gold annotation of the 30 "Inter-user contouring accuracy" test sets seem to have been performed by senior physicians according to international standards while the annotations used for inter-rater variation were performed by a less experienced physician (4y) and followed a different annotation protocol (that of FAH-ZU). It is thus not at all surprising that SOARS would show superior agreement with the gold annotation because it was trained using exactly the protocol that was also used by the gold annotations.

4) Large parts of the evaluation relies on only very little data. The entire sections about editing efforts, inter-user contouring accuracy and dosimetry rely on a very small subset of the data: 30 cases from FAH-ZU. This substantially weakens these parts of the manuscript. I understand that manual annotations of these cases is expensive, but given that 176 cases could be annotated for training it would have been nice to see a larger testing dataset. Furthermore: why do all these cases come from a single institution? Balancing those cases across the multiple testing sides would also greatly improve that contribution.

5) The data splits used to train the NAS part of the model are difficult to understand and not clearly described. This is highly relevant because gains through NAS are reported in the ablation study and it is unclear how the validation set referenced in this table has been used. Please clearly

lay out the training and evaluation scheme during method development including the use of the validation set used for Table 2

6) There are a couple of minor issues that can easily be resolved:

- I am confused by the architecture described in "Detailed segmentation baseline and auto-searched network architecture". This is not the standard nnU-Net which is cited as the baseline method used in Table 2 (the default nnU-Net does not use residual connections; downsampling is implemented as strided convolution and upsampling is implemented as convolution transposed). Neither is anything mentioned about residual connections in the remainder of the article. Can you please clarify?

- "The codes used for inference and result performance evaluation can be publicly available on GitHub after publication." In the spirit of open science I encourage the authors to publish the code for their method.

- line 31: "due to the predictable prohibitive labor-cost" -> predictably?

- line 262: "delineate 42 H&N OARs as the current most comprehensive clinical protocol" that sentence seems a bit off

- line 408: "The searched network architectures for each branch are listed in Fig. 3. The implementation details are

410 reported in the supplementary materials." -> this should probably be Suppl. Fig 1

Reviewer #2:

Remarks to the Author:

General Comments

AI-based autosegmentation is a hot topic in radiation oncology these days, with many research sites and commercial entities developing software to segment the human body with AI/deep-learning strategies. The primary goals are to increase the efficiency and, perhaps more importantly, to drive out inter-observer variation in what is a practice based more on training and "opinion" rather than an exact science.

The most noteworthy achievements of this work compared to similar works are (1) the number of organs-at-risk (OAR) generated and (2) the amount of data collected and analyzed. The results show incremental improvements over other methods.

Specific Comments

30-33, 48-64: This implies the main problem is the volume of work to be done by humans, i.e., the number of OARs to be contoured. This is a practical problem, yes, but the main clinical problem is the large variation in opinions across many trained observers, each who considers himself or herself an "expert." The big win of accurate autosegmentation will be not just automation, but the driving out of variation.

94: The number (42) of OARs is indeed impressive and one of the novel aspects of this work compared to the many commercial systems that are becoming available today from both the major vendors and small start-ups. However, the term "state-of-the-art" is qualitative and sounds like an advertisement. A better statement would be a more neutral and precise, "SOARS produces segmentation of a large number (42) of OARs, but ..."

120-125: Is being a "senior physician" the only qualifier for data serving as an input? What defines "senior?" (Years of experience? Contouring assessments/testing? Other?) Given that there is probably a lot of inter-observer variation even among senior physicians, what is the critical number of datasets that you feel is necessary to build a strong consensus and factor out outliers?

137-149: 1327 "unseen" datasets ... why do we expect those are correct? Matching human outputs does not imply accuracy unless the human outputs have been fully vetted via consensus of actual experts.,

162-167 (and 442-461): Correct way to get dosimetric impact is to produce highly conformal (e.g., VMAT or proton) plans based on each contouring permutation, then overly a gold standard atop each to compare "what was planned and reviewed" vs. "what was the true DVH."

162-167 (and 442-461): The "Gold" dataset is considered the baseline. Did those physicians' contours (and/or the contours from these dosimetric datasets) get included in the training data? If so, then this is just proving that SOARS reproduces what it's trained to reproduce. What this section is really is a sensitivity analysis with dosimetric endpoints which are far more relevant in radiation therapy planning than DSC, HD, and ASD.

137-149: 1327 "unseen" datasets ... why do we expect those are correct? Matching human outputs does not imply accuracy unless the human outputs have been fully vetted via consensus of actual experts.

176-177: The phrase "significantly worse" here is assumed to mean statistically significant, but it sounds like the authors are implying the magnitude of the differences are significant, when this seems not so (DICE: 69.8% vs 75.3%, HD: 8.8 vs 7.9mm, ASD: 1.6 vs 1.1mm). This should be rephrased with less hyperbole, such as: "... had inferior performance that was statistically significant ...".

190: Has "HD95" been defined prior to this usage? HD was defined (line 171). It is assumed that HD means the maximum surface difference distance and HD95 would be the distance at which 95% of the points match ($HD95 < HD$). Both of these concepts are simple/intuitive and should be explained in detail.

207-212: Is there an observer bias when viewing existing contours and determining if they require editing, and if so, how much? I'd be interested to see a side study of physicians presented (blindly) with copies of their original contours that they drew themselves and asked if those require any editing, with the expected result being that 100% require no editing. Then, present them with contours that are different from their original opinions, where the expectation would be that 0% require no editing and 100% require minor or major edits.

226: Here you show the DICE coefficient (DSC) as a fraction ranging from 0.00 to 1.00, where elsewhere you show a percentage. I think the usual formalism is a fraction, not a percentage but regardless, choose one and be consistent throughout the paper.

226: Again, the phrase "significantly improved" is a bit misleading. The changes may be statistically significant, but the absolute changes are not that large. See the prior suggestion regarding lines 176-177 and employ a similar, precise re-phrasing.

Question: All these patients were from China. HN anatomy (unlike GI or pelvis) should not vary much if at all depending on nationality, but has there been any testing for other cohorts of patients outside of China? Do you expect models need to be trained based on different regions or clinical practice/protocol? If so, what is the ideal and minimal number of patient datasets required and how long does that process take?

Suggestion: Comment on the application of this specific method to other body sites.

Question: Did you consider having professional anatomists/radiologists review, because radiation oncologists are not necessarily the best source of anatomical knowledge. (There's a reason they vary in the quality of their contouring outputs and often draw only a subset of anatomical structures.)

Suggestion: Did you consider validation using multi-modality, fused CT-MR image sets where the fused MRI offers more anatomical info on soft tissue organs? Thus, the "gold standard" references could be human experts contouring on CT-MR datasets whereas SOARS is using only CT. That would be an important and significant addition to this paper.

Response to reviewers

We are grateful for the constructive comments on our manuscript “Comprehensive and Clinically Accurate Head and Neck Organs at Risk Delineation via Stratified Deep Learning: A Large-scale Multi-Institutional Study” (NCOMMS-21-39775). By taking them into account, we have significantly strengthened the manuscript’s quality, e.g., we add new experiments using two public data, expand the multi-user testing set to 50 patients from two institutions, conduct new blind user study to assess the OAR editing efforts, and carry out enhanced clinical dosimetric evaluation. A point-by-point response is provided below and a list of major changes is attached at the end of this letter.

REVIEWER #1

Comment: The paper describes the application of a deep-learning based segmentation pipeline to a large cohort of head and neck cancer patients. Segmentation task are the 42 OARs that are recommended by international guidelines. The proposed method stratifies these OARs into different subcategories based on size and difficulty and uses a progressive segmentation scheme where predictions of previous steps are incorporated into subsequent ones. Notable is the localization and cropping of surrounding regions of difficult structures to aid the model perform the segmentation accurately. In addition to the stratification, the authors propose to use differentiable neural architecture search (NAS) to find optimal architectures for each of the subcategories.

The proposed model is evaluated extensively and thoroughly: while the training set originates from only a single-institutional cohort, the authors demonstrate that their model generates well to patients from multiple external testing datasets. Moreover, the authors carefully evaluate the segmentations with respect to clinical impact in the form of needed revisions and dosimetry. A comparison with the inter-rater variability underlines the strong results.

Overall the paper is very well written and clearly structured. Most of the needed information for reproducing the algorithm can be found in the Online Methods as well as the Supplement.

Response: We thank Reviewer 1 for the positive assessment of the writing, methodology, experimental design, performance, and clinical impact. Please see below our responses to detailed questions.

Q1: The overwhelming majority of the methodological contributions seems to already have been published as a CVPR paper. The difference to the method presented here is mainly the backbone which was swapped in favor of a U-Net. The other key features of the approach remain the same (stratification, NAS). Thus, the methodological contribution offered by this paper seems limited and the value of the paper mainly lies in its extensive clinically-oriented evaluation. I think the messaging about the key contributions may need rewording.

Response: Thank you for this question. We first published our preliminary methodological contributions in a CVPR2020 conference paper [1] using a subset (142) of the internal patients to explore our method’s technical feasibility. This work should be seen as an extended journal version of the conference paper. As per the Nature Communications editorial policy, “The Nature journals are happy to consider submissions containing material that has been published in a conference proceedings paper. However, the submission should provide a substantial extension of results, methodology, analysis,

conclusions and/or implications over the conference proceedings paper; the final decision on what constitutes a substantial extension is made by the editors at each individual journal. In this work, we have described our previous CVPR work at lines 78-93 in the original manuscript (the CVPR paper is also attached as an additional material in the initial submission). In the current manuscript, we focus on answering essential questions regarding clinical applicability and generality (lines 96-100 in the original manuscript), which is a substantial extension regarding analysis, conclusions, and implications over the previous conference proceeding. In particular, we first enhanced SOARS by replacing the segmentation backbone of P-HNN [2] with UNet [3] and conducted new NAS optimization based on the Unet architecture. Then, we designed and carried out highly comprehensive new internal and external experiments, and multi-user subjective and quantitative evaluation studies that directly validated clinical utility and reliability. To our knowledge, such an effort has never been done for a deep learning-based OAR segmentation. Key sections of Results and Discussion are also completely new and have significant scientific and clinical values, insights, and merits.

Q2: Proposing new methods using only internal datasets is in my opinion no longer sufficient. The paper would greatly benefit from using publicly available datasets for evaluation, for example the MICCAI 2015 challenge dataset (which was also used by the authors in their CVPR publication) and the StructSeg2019 dataset. Unfortunately the website of the latter seems broken at the moment but it should be possible to contact the organizers nonetheless. Independent evaluation on publicly available datasets allows readers (and reviewers) to directly compare the proposed algorithm with the existing literature and thus get a much clearer picture of their capabilities.

Response: Thank you very much for this suggestion. We have added experimental results on two public HN OAR datasets (MICCAI 2015 dataset and StructSeg2019 dataset). See Tables I and II in this letter (suppl. Table 6 and 7 in the revised version). For the MICCAI 2015 dataset, we trained our model using the provided 33 training cases and evaluated the performance on 15 testing cases. An 83.6% Dice similarity coefficient (DSC) is achieved when averaged over 9 OARs, improving over the previous leading method UaNet [4] (published in Nature Machine Intelligence 2019) by 2.4% absolute DSC. For the StructSeg 2019 dataset, we attempted on multiple occasions to contact the organizers for the testing set evaluation, but did not receive any response. Therefore, we decided to conduct a 5-fold cross-validation comparing SOARS, UaNet [4] and nnUNet [5] on the public training set. Specifically, 50 patients (provided by the organizers for training) were randomly partitioned into five equal-sized subgroups (20% of patients per fold). Of the five subgroups, a single subgroup was retained as the test set, while the remaining four subgroups (80% of patients) were used for training and validation. The cross-validation process was repeated five times/folds, with each of the five subgroups used once as the unseen testing data. SOARS achieves an average DSC and Hausdorff distance (HD) of 80.9% and 5.8mm, respectively. In comparison, nnUNet has an average DSC and HD of 79.2% and 6.7mm, respectively; and UaNet an average DSC and HD of 78.6% and 6.6mm, respectively. We have added the evaluation on public datasets into the revised version.

Table I. Dice similarity coefficient (%) comparison with previous published results on MICCAI2015 testing dataset. SOARS NC achieves 8 (in bold) best performance and 1 (in grey box) second best performance.

	Brainstem	Mandible	Optic Chiasm	Optic Nerve		Parotid		SMG		AVG.
				left	right	left	right	left	right	
Ren et al. [6]	-	-	58.0 ± 17.0	72.0 ± 8.0	70.0 ± 9.0	-	-	-	-	-
Wang, et al. [7]	90.0 ± 4.0	94.0 ± 1.0	-	-	-	83.0 ± 6.0	83.0 ± 6.0	-	-	-
Nikolov et al. [8]	79.5 ± 7.8	94.0 ± 2.0	-	71.6 ± 5.8	69.7 ± 7.1	86.7 ± 2.8	85.3 ± 6.2	76.0 ± 8.9	77.9 ± 7.4	-
Tong et al. [9]	87.0 ± 3.0	93.7 ± 1.2	58.4 ± 10.3	65.3 ± 5.8	68.9 ± 4.7	83.5 ± 2.3	83.2 ± 1.4	75.5 ± 6.5	81.3 ± 6.5	77.4
Harrison et al [2]	87.2 ± 2.5	93.1 ± 1.8	55.6 ± 14.1	72.6 ± 4.6	71.2 ± 4.4	87.7 ± 1.8	87.8 ± 2.3	80.6 ± 5.5	80.7 ± 6.1	79.6
AnatomyNet [10]	86.7 ± 2.0	92.5 ± 2.0	53.2 ± 15.0	72.1 ± 6.0	70.6 ± 10	88.1 ± 2.0	87.3 ± 4.0	81.4 ± 4.0	81.3 ± 4.0	79.2
FocusNet [11]	87.5 ± 2.6	93.5 ± 1.9	59.6 ± 18.1	73.5 ± 9.6	74.4 ± 7.2	86.3 ± 3.6	87.9 ± 3.1	79.8 ± 8.1	80.1 ± 6.1	80.3
UaNet [4]	87.5 ± 2.5	95.0 ± 0.8	61.5 ± 10.2	74.8 ± 7.1	72.3 ± 5.9	88.7 ± 1.9	87.5 ± 5.0	82.3 ± 5.2	81.5 ± 4.5	81.2
SOARS CVPR [1]	87.6 ± 2.8	95.1 ± 1.1	64.9 ± 8.8	75.3 ± 7.1	74.6 ± 5.2	88.2 ± 3.2	88.2 ± 5.2	84.2 ± 7.3	83.8 ± 6.9	82.4
SOARS NC	88.6 ± 2.7	96.6 ± 0.8	69.2 ± 9.8	75.8 ± 6.1	75.2 ± 4.8	88.9 ± 2.2	88.6 ± 4.8	84.5 ± 6.9	85.1 ± 5.8	83.6

Table II. Quantitative results on the StructSeg 2019 dataset using 5-fold cross-validation evaluation. Bold values represent the best performance.

OARs	UaNet		nnUNet		SOARS	
	DSC	HD (mm)	DSC	HD (mm)	DSC	HD (mm)
BrainStem	85.3% ± 4.7%	6.6 ± 2.1	87.1% ± 4.1%	6.1 ± 2.3	87.7% ± 3.6%	5.7 ± 2.1
Eye_Lt	88.2% ± 5.2%	3.7 ± 1.3	89.5% ± 2.7%	3.7 ± 0.8	89.2% ± 2.8%	3.4 ± 0.6
Eye_Rt	88.3% ± 3.6%	3.9 ± 1.2	89.1% ± 2.5%	3.6 ± 1.0	88.9% ± 2.7%	3.5 ± 0.9
InnerEar_Lt	83.4% ± 4.2%	4.1 ± 1.5	82.8% ± 4.0%	4.3 ± 2.3	86.4% ± 3.7%	4.2 ± 1.4
InnerEar_Rt	83.4% ± 6.4%	4.0 ± 1.3	83.0% ± 5.2%	4.5 ± 2.9	86.5% ± 4.8%	4.1 ± 1.1
Lens_Lt	72.1% ± 10.1%	2.8 ± 1.3	74.5% ± 7.9%	2.8 ± 1.1	75.8% ± 8.6%	2.8 ± 0.7
Lens_Rt	71.0% ± 11.6%	2.9 ± 1.0	72.7% ± 10.1%	2.6 ± 0.8	75.4% ± 8.7%	2.9 ± 0.8
Mandible_Lt	91.0% ± 2.2%	8.7 ± 4.5	91.0% ± 2.0%	8.2 ± 2.4	91.0% ± 2.0%	6.8 ± 2.3
Mandible_Rt	90.7% ± 3.1%	9.1 ± 7.3	91.1% ± 2.1%	9.2 ± 10.8	91.2% ± 2.0%	7.6 ± 3.0
MidEar_Lt	79.1% ± 9.1%	9.3 ± 4.7	80.1% ± 9.5%	8.7 ± 4.2	81.5% ± 5.8%	7.8 ± 4.5
MidEar_Rt	78.2% ± 9.1%	8.1 ± 4.4	80.0% ± 8.8%	7.1 ± 3.0	81.6% ± 6.2%	6.6 ± 3.6
OpticChiasm	55.9% ± 12.1%	6.9 ± 2.5	53.5% ± 13.1%	6.9 ± 2.4	61.2% ± 11.2%	4.8 ± 1.8
OpticNerve_Lt	67.0% ± 9.3%	4.6 ± 1.7	67.6% ± 8.8%	4.6 ± 1.6	71.7% ± 10.4%	2.7 ± 1.0
OpticNerve_Rt	66.4% ± 11.1%	5.0 ± 2.4	66.8% ± 10.3%	4.6 ± 2.0	70.4% ± 8.7%	2.6 ± 0.9
Parotid_Lt	84.2% ± 5.9%	10.9 ± 4.9	85.6% ± 3.9%	11.3 ± 4.7	85.8% ± 3.5%	9.8 ± 3.7
Parotid_Rt	83.9% ± 5.6%	12.2 ± 6.4	85.7% ± 3.5%	12.2 ± 6.2	85.8% ± 3.4%	11.2 ± 5.8
Pituitary	62.0% ± 14.2%	3.7 ± 0.9	59.9% ± 16.9%	4.1 ± 1.3	64.2% ± 15.4%	3.6 ± 1.2
SpinalCord	81.3% ± 7.9%	4.6 ± 1.7	82.6% ± 3.4%	4.4 ± 1.9	83.2% ± 3.2%	4.4 ± 2.0
TempLobe_Lt	85.1% ± 5.5%	12.0 ± 4.8	86.6% ± 4.7%	12.8 ± 5.2	86.8% ± 4.7%	11.1 ± 3.7
TempLobe_Rt	85.2% ± 5.5%	11.8 ± 4.8	85.8% ± 4.5%	13.7 ± 4.0	86.1% ± 4.4%	12.9 ± 6.1
TMJ_Lt	72.6% ± 10.4%	5.6 ± 2.6	74.0 ± 10.4%	5.7 ± 2.9	74.7 ± 6.6%	5.0 ± 1.3
TMJ_Rt	73.9% ± 9.0%	5.4 ± 2.5	73.0 ± 8.5%	5.4 ± 2.7	75.6 ± 5.8%	4.6 ± 1.3
Average	78.6%	6.6	79.2%	6.7	80.9%	5.8

Q3: There seem to be biases in the annotations that favor SOARS over the ‘human reader’ in the inter-rater disagreement studies. Concretely, both the internal training dataset as well as the gold annotation of the 30 “Inter-user contouring accuracy” test sets seem to have been performed by senior physicians according to international standards while the annotations used for inter-rater variation were performed by a less experienced physician (4y) and followed a different annotation protocol (that of FAH-ZU). It is thus not at all surprising that SOARS would show superior agreement with the gold annotation because it was trained using exactly the protocol that was also used by the gold annotations.

Response: Thanks for the question. We’d like to formally clarify that in the multi-user contouring study, the less experienced physician also annotated the 13 OARs according to the same international annotation standard [14] as the senior physicians. By saying that “the less experienced physician followed the FAH-ZU protocol,” we only mean that 13 OARs were annotated instead of 42, which is the number of OARs difference (13 vs 42). There is no annotation standard deviations. We have made this point clearer in the revised version (line 180-182).

Q4: Large parts of the evaluation relies on only very little data. The entire sections about editing efforts, inter-user contouring accuracy and dosimetry rely on a very small subset of the data: 30 cases from FAH-ZU. This substantially weakens these parts of the manuscript. I understand that manual annotations of these cases is expensive, but given that 176 cases could be annotated for training it would have been nice to see a larger testing dataset. Furthermore: why do all these cases come from a single institution? Balancing those cases across the multiple testing sites would also greatly improve that contribution

Response: Thank you for this excellent suggestion. We have expanded our multi-user testing dataset (30 patients of FAH-ZU) by further including 20 patients randomly selected from SMU so that the total multi-user patient number reaches 50. This number is at least more than twice of that used in recent segmentation user-studies, e.g., 10 patients to evaluate the prostate gland segmentation performance among difference human readers [12] (*International Journal of Radiation Oncology*Biophysics*, 2019); 20 patients to examine the GTV segmentation accuracy by human readers [13] (*Radiology*, 2019); and 20 patients to examine the human reader performance of OARs [4] (*Nature Machine Intelligence*, 2019). Using the 50 multi-user patients from multiple clinical sites as reviewer suggested, new quantitative results of editing efforts, inter-user contouring accuracy and direct dosimetric accuracy have been updated in Figure 1 and Table III in this letter and Figure 4-5 and Table 5 in the revised version. As can be seen, similar results are observed. Hence, the results are considerably strengthened by demonstrating the effectiveness, generality and robustness of SOARS across different institutions.

Table III. Quantitative contouring accuracy and direct dosimetric accuracy ($Diff_{\text{mean dose}}^{\text{direct}}$ and $Diff_{\text{max dose}}^{\text{direct}}$) comparison between SOARS, SOARS-revised and the human reader on the multi-user testing dataset of 50 patient. DSC, HD and ASD represent Dice similarity coefficient, Hausdorff distance, and average surface distance, respectively. Direct mean dose differences $Diff_{\text{mean dose}}^{\text{direct}}$ and direct maximum dose differences $Diff_{\text{max dose}}^{\text{direct}}$ are calculated using the equation (6) and (7), respectively. DSC higher the better, while HD, $Diff_{\text{mean dose}}^{\text{direct}}$ and $Diff_{\text{max dose}}^{\text{direct}}$ lower the better. Wilcoxon matched-pairs signed rank test is used to compare between SOARS/SOARS-revised results and human reader's results, and bold and highlighted values represent the best performance scores and statistically significant improvements.

Segmentation accuracy						
OARs	human reader		SOARS		SOARS-revised	
	DSC	HD (mm)	DSC	HD (mm)	DSC	HD (mm)
BrainStem	84.3% ± 5.0%	7.4 ± 3.6	87.0% ± 3.3%	5.7 ± 1.6	87.9% ± 3.3%	5.1 ± 1.6
Eye_Lt	89.5% ± 3.3%	3.4 ± 0.7	90.0% ± 2.6%	3.3 ± 0.7	89.9% ± 2.7%	3.2 ± 0.7
Eye_Rt	88.2% ± 5.7%	3.6 ± 1.0	89.3% ± 4.3%	3.3 ± 0.5	89.1% ± 4.4%	3.3 ± 0.6
Lens_Lt	74.7% ± 9.1%	2.5 ± 0.8	74.2% ± 7.6%	2.2 ± 0.6	75.9% ± 7.7%	2.2 ± 0.6
Lens_Rt	72.2% ± 9.7%	2.7 ± 0.8	75.4% ± 6.0%	2.2 ± 0.6	76.8% ± 5.7%	2.1 ± 0.6
OpticChiasm	67.6% ± 15.4%	5.1 ± 1.7	74.2% ± 8.9%	4.3 ± 1.0	77.7% ± 9.5%	4.2 ± 1.2
OpticNerve_Lt	65.3% ± 9.8%	8.2 ± 3.6	73.2% ± 7.4%	4.4 ± 1.9	74.8% ± 6.8%	3.7 ± 1.3
OpticNerve_Rt	64.9% ± 12.2%	7.5 ± 4.5	72.1% ± 7.1%	4.1 ± 1.6	73.4% ± 7.9%	3.9 ± 1.5
Parotid_Lt	83.1% ± 4.6%	12.6 ± 5.3	88.2% ± 3.4%	7.6 ± 2.8	88.3% ± 3.3%	7.6 ± 2.8
Parotid_Rt	82.7% ± 5.2%	12.3 ± 5.5	87.8% ± 3.6%	7.5 ± 2.9	87.9% ± 3.5%	7.2 ± 2.0
SpinalCord	81.7% ± 6.1%	9.3 ± 8.1	83.7% ± 3.0%	4.4 ± 1.3	83.9% ± 3.0%	3.8 ± 0.7
TMJ_Lt	74.0% ± 9.9%	3.5 ± 0.9	79.0% ± 8.6%	3.2 ± 0.8	82.1% ± 9.9%	2.8 ± 0.6
TMJ_Rt	73.6% ± 12.7%	3.5 ± 1.3	77.6% ± 7.3%	3.4 ± 0.6	81.1% ± 10.5%	2.9 ± 0.6
Average	77.1%	6.3	80.9%	4.3	82.2%	4.0
Dosimetric accuracy ($Diff_{\text{mean dose}}^{\text{direct}}$ and $Diff_{\text{max dose}}^{\text{direct}}$)						
OARs	human reader		SOARS		SOARS-revised	
	diff in mean dose	diff in max dose	diff in mean dose	diff in max dose	diff in mean dose	diff in max dose
BrainStem	4.7% ± 5.1%	5.1% ± 6.7%	2.8% ± 2.8%	4.3% ± 4.1%	3.1% ± 3.1%	4.1% ± 4.2%
Eye_Lt	3.8% ± 3.4%	6.9% ± 7.5%	4.1% ± 3.0%	5.9% ± 4.7%	4.4% ± 2.9%	4.7% ± 4.3%
Eye_Rt	5.4% ± 5.2%	6.6% ± 5.7%	5.1% ± 4.0%	6.2% ± 5.0%	4.9% ± 3.9%	5.8% ± 5.0%
Lens_Lt	2.2% ± 2.7%	2.8% ± 2.9%	1.8% ± 2.2%	2.7% ± 2.7%	1.5% ± 1.4%	2.4% ± 2.4%
Lens_Rt	3.5% ± 4.2%	4.8% ± 5.6%	2.6% ± 3.3%	3.8% ± 5.8%	2.5% ± 3.2%	3.5% ± 5.7%
OpticChiasm	8.2% ± 11.5%	5.9% ± 9.6%	5.1% ± 6.3%	3.3% ± 7.0%	4.9% ± 8.0%	3.8% ± 7.3%
OpticNerve_Lt	13.0% ± 11.2%	7.2% ± 9.5%	9.0% ± 9.4%	4.0% ± 6.0%	9.2% ± 8.8%	3.5% ± 7.4%
OpticNerve_Rt	11.7% ± 10.9%	6.9% ± 9.4%	9.8% ± 10.1%	3.7% ± 4.9%	10.9% ± 10.4%	3.9% ± 6.1%
Parotid_Lt	4.8% ± 4.8%	2.1% ± 2.5%	2.6% ± 3.3%	1.4% ± 1.6%	2.5% ± 3.2%	1.5% ± 1.7%
Parotid_Rt	4.8% ± 5.3%	2.0% ± 2.2%	2.8% ± 3.5%	1.2% ± 1.9%	2.8% ± 3.5%	1.2% ± 1.9%
SpinalCord	10.2% ± 12.7%	3.3% ± 5.4%	2.9% ± 4.4%	1.7% ± 2.3%	2.7% ± 4.2%	1.9% ± 2.3%
TMJ_Lt	2.6% ± 2.4%	2.7% ± 2.9%	3.0% ± 2.5%	3.1% ± 2.6%	2.6% ± 3.1%	3.2% ± 3.0%
TMJ_Rt	2.5% ± 3.0%	2.5% ± 2.7%	2.8% ± 1.9%	1.9% ± 1.9%	2.7% ± 2.9%	2.0% ± 2.1%
Average	6.0%	4.4%	4.2%	3.3%	4.2%	3.2%

Note: diff in mean dose and diff in max dose represent the difference in mean dose and difference in maximum dose, respectively.

Figure 1. Summary of human experts’ assessment of revision effort on SOARS predicted 42 OARs. Anchor, mid-level and S&H OAR categories are shown separately. Vast majority of SOARS predicted OARs only required minor revision or no revision from expert’s editing before they can be clinically accepted. Only a very small amount of OARs need moderate level revision and no OARs need major revision efforts. Minor revision: editing required in <1 minute; moderate revision: editing required in 1–3 minutes; and major revision: editing required in >3 minutes.

Q5: The data splits used to train the NAS part of the model are difficult to understand and not clearly described. This is highly relevant because gains through NAS are reported in the ablation study and it is unclear how the validation set referenced in this table has been used. Please clearly lay out the training and evaluation scheme during method development including the use of the validation set used for Table 2.

Response: Thanks for this good question and we apologize for this confusion. We have added more details of data splits regarding the NAS training and the ablation study in the revised version. “We divide this Training-Validation Dataset (176 from CGMH) into two subgroups: 80% to train and validate the segmentation model and 20% as a held-out test set to evaluate the ablation performance. To avoid biases in selection of the learnable logits α_k when training NAS, we use a larger proportion of patients as validation than is typical, i.e., a validation/training ratio of 1:2 for NAS. Therefore, when considering all 176 patients, **the NAS training procedure uses 53% for training, 27% for validation, and 20% for ablation-testing (never seen in NAS training)**. After finalizing the network architecture by the NAS procedure, we retrain the model from scratch using only the searched architecture and set the validation/training sizes to a more typical ratio: **64% for training, 16% for validation, and 20% for ablation-testing**. More importantly, please note that the ablation-testing cases (20% of the Training-Validation dataset) were never seen in the NAS training and validation process.

Q6: There are a couple of minor issues that can easily be resolved:

Q6.1: I am confused by the architecture described in “Detailed segmentation baseline and auto-searched network architecture”. This is not the standard nnU-Net which is cited as the baseline method used in Table 2 (the default nnU-Net does not use residual connections; downsampling is implemented as strided convolution and upsampling is implemented as convolution transposed). Neither is anything mentioned about residual connections in the remainder of the article. Can you please clarify?

Response: Thank you for the question. We used the residual UNet structure implemented in nnUNet’s github repository “nnunet/network_architecture/generic_modular_residual_UNet.py”, which is indeed different from the one adopted in the original nnUNet paper (the plain UNet). Since adding residual connection leads to comparable computational cost and normally benefits the segmentation performance as observed by previous work and consistent with our experience, we choose this nnUNet variation. For downsampling, we have set the parameter to choose the max pooling instead of the default strided convolution in nnUNet’s implementation, and for upsampling, we have used the default transposed convolution. We have made this clearer in the revised supplementary materials.

Q6.2: “The codes used for inference and result performance evaluation can be publicly available on GitHub after publication.” In the spirit of open science I encourage the authors to publish the code for their method.

Response: Thank you for this suggestion. We agree with the spirit of open science. However, due to our company’s policy, we are unable to publish the training codes at this stage yet. We will make the inference and evaluation codes and pretrained models available for the public once the work is accepted.

Q6.3: line 31: “due to the predictable prohibitive labor-cost” -> predictably?

Response: Thanks for pointing this out. We have corrected it in the revised version.

Q6.4: line 262: “delineate 42 H&N OARs as the current most comprehensive clinical protocol” that sentence seems a bit off.

Response: Thanks for pointing this out. We have changed this sentence to “delineate 42 H&N OARs following the most comprehensive clinical protocol” in the revised version.

Q6.5: line 408: “The searched network architectures for each branch are listed in Fig. 3. The implementation details are reported in the supplementary materials.” -> this should probably be Suppl. Fig 1.

Response: Thanks for pointing this out. Yes, it should be Suppl. Fig 1. We have corrected it in the revised version.

REVIEWER #2

Comment: AI-based autosegmentation is a hot topic in radiation oncology these days, with many research sites and commercial entities developing software to segment the human body with AI/deep-learning strategies. The primary goals are to increase the efficiency and, perhaps more importantly, to drive out inter-observer variation in what is a practice based more on training and “opinion” rather than an exact science.

The most noteworthy achievements of this work compared to similar works are (1) the number of organs-at-risk (OAR) generated and (2) the amount of data collected and analyzed. The results show incremental improvements over other methods.

Response: Thank Reviewer 2 for recognizing our noteworthy contributions, such as the most comprehensive OARs segmented, and the large scale of datasets collected and analyzed. We’d also like to note other key contributions in this work: (1) based on observations and inspirations of how human readers perform this task, we proposed a novel stratified OAR segmentation (SOARS) deep learning framework to effectively segment a large set of 42 head and neck OARs. SOARS stratifies 42 OARs into the “anchor”, “mid-level”, and “small & hard” three categories, with automatically derived neural network architectures per organ category using neural architecture search (NAS). This stratified deep learning scheme successfully decomposed this hard-to-solve (directly segmenting 42 OARs in the planning CT) problem in a “divide and conquer” manner and illustrated how deep learning models may be designed to simulate the human reader’s conceptual thinking and visual parsing processes. (2) we designed and conducted three user studies to evaluate the clinical usefulness and impact, such as the human editing efforts and human reader contouring variation. To the best of our knowledge, we are also the first study to examine the dosimetric performance in the downstream RT planning stage when adopting the deep learning generated OAR contours.

For the improvement of our method as compared to other works, we note that most of our quantitative results show statistically significant improvement over the previous leading method such as UaNet [4], which was proposed in *Nature Machine Intelligence* 2019. (1) In the internal evaluation of 326 CGMH patients, SOARS achieves an average of 74.8% Dice similarity coefficient (DSC) and 1.2mm average surface distance (ASD) across 42 OARs, which are statistically significantly higher than that achieved by the UaNet with 5% absolute DSC improvement and 20% relative ASD error reduction. (2) In the external multi-institutional quantitative evaluation of 1001 patients, SOARS achieves an average of 78.0% DSC and 1.0mm ASD across 25 OARs, which are again statistically significantly higher than that obtained from the UaNet (with ~4% absolute DSC improvement and 29% relative ASD error reduction). (3) Upon Reviewer 1’s suggestion, we further conducted additional experimental evaluations on the public HN OAR datasets, i.e., the MICCAI2015 and StructSeg2019 dataset. In StructSeg2019 dataset, SOARS achieves an average of 80.9% DSC and 0.8mm ASD across 22 OARs, which are statistically significantly higher than that achieved by UaNet (with 2.3% absolute DSC improvement and 23% relative ASD error reduction).

Q1: 30-33, 48-64: This implies the main problem is the volume of work to be done by humans, i.e., the number of OARs to be contoured. This is a practical problem, yes, but the main clinical problem is the large variation in opinions across many trained observers, each who considers himself or herself an

“expert.” The big win of accurate autosegmentation will be not just automation, but the driving out of variation.

Response: Thank you, we truly appreciate your insightful comments. Yes, although there exists the international delineation consensus recommendation of OARs [14] and many OARs do yield clear or observable boundaries in CT scans, physicians often just follow their trained experience instead of following the guidelines. We agree that the great benefit of the practice of automated OAR segmentation also lies in the standardization of the manual contouring (if not more). We have emphasized this point in the Introduction of the revised version (line 65-68).

Q2: 94: The number (42) of OARs is indeed impressive and one of the novel aspects of this work compared to the many commercial systems that are becoming available today from both the major vendors and small start-ups. However, the term “state-of-the-art” is qualitative and sounds like an advertisement. A better statement would be a more neutral and precise, “SOARS produces segmentation of a large number (42) of OARs, but ...”

Response: Thanks for this comment. We have revised the term “state-of-the-art” to a more neutral word. We revised this sentence to “SOARS segments a large number (42) of OARs with leading performance in a single institution cross-validation evaluation, but ...”. Please see revised manuscript at line 99.

Q3.1: 120-125: Is being a “senior physician” the only qualifier for data serving as an input? What defines “senior?” (Years of experience? Contouring assessments/testing? Other?) Given that there is probably a lot of inter-observer variation even among senior physicians, what is the critical number of datasets that you feel is necessary to build a strong consensus and factor out outliers?

Response: Thank you for this question. A senior physician in our study is not only required to have experience in the head & neck specialty for at least 10 years with 100-300 annually treated patients, but also they are very familiar with and follow the delineation consensus guidelines [14] in their clinical practice with high fidelity. We did not build the delineation consensus by ourselves (we agree that there is inter-observer variation even among senior physicians). Instead, we make sure that senior physicians follow the delineation consensus guidelines [14] to aim for consistent annotation. We describe this clearer in the revised version (line 132-135).

Q3.2: 137-149: 1327 “unseen” datasets ... why do we expect those are correct? Matching human outputs does not imply accuracy unless the human outputs have been fully vetted via consensus of actual experts.

Response: Thank you for this question. Yes, senior physicians (defined in the above comment) have examined the manual OAR annotations in unseen datasets to ensure quality. As mentioned above, senior physicians in our study follow international delineation guideline [14]. Indeed, two steps of examinations were conducted. First, senior physicians of each institution first examined and edited the manual OAR contours of the data from their own institution. Moreover, three most senior physicians (C.

Lin, X. Ye and J. Ge) further examined all cases. If cases in an institution were found deviating from the delineation guideline [14], modification suggestions were provided to senior physicians of said institution for confirmation and editing. We apologize for the unclear description of this consensus process and have added these details in the revised version (line 161-167).

Q4: 162-167 (and 442-461): Correct way to get dosimetric impact is to produce highly conformal (e.g., VMAT or proton) plans based on each contouring permutation, then overlay a gold standard atop each to compare “what was planned and reviewed” vs. “what was the true DVH.”

Response: Thank you for this suggestion. We agree that a more clinically relevant method to evaluate the dosimetric impact is to generate the RT plans based on each OAR contouring permutations and then, overlay the gold standard OARs on top of each to compare the dosimetric differences (see equation I and II in this letter and equation (8) and (9) in the revised version).

$$\text{Diff}_{\text{mean dose}}^{\text{clinic}} = \frac{\text{mean dose}(OAR_{\text{substitute}}, Dose_{\text{substitute}}) - \text{mean dose}(OAR_{\text{ref}}, Dose_{\text{substitute}})}{\text{mean dose}(OAR_{\text{ref}}, Dose_{\text{substitute}})} \times 100\% \quad (\text{I})$$

$$\text{Diff}_{\text{max dose}}^{\text{clinic}} = \frac{\text{max dose}(OAR_{\text{substitute}}, Dose_{\text{substitute}}) - \text{max dose}(OAR_{\text{ref}}, Dose_{\text{substitute}})}{\text{max dose}(OAR_{\text{ref}}, Dose_{\text{substitute}})} \times 100\% \quad (\text{II})$$

Following your suggestion, we randomly chose 10 out of 50 patients for this purpose (in the revised multi-user study, we now have 50 patients as suggested by Reviewer 1). We did not include all 50 multi-user patients, because the effort is too costly at this stage in both time and efforts: each patient has 3 OAR contour permutations (human reader, SOARS, SOARS-revised). In total, it would require 150 (50x3) new IMRT planning for this user study, in which itself could be a separate clinical paper to mainly discuss the dosimetric effects in more detail. Hence, we make a tradeoff to randomly select 10 patients for the replanning to illustrate the preliminary performance. Results are shown in Table IV and Figure II of this letter (Suppl. Table 8 and Suppl. Figure 4-5 in the revised version). After replanning, SOARS and SOARS-revised contours have slightly smaller $\text{Diff}_{\text{max dose}}^{\text{clinic}}$ and $\text{Diff}_{\text{mean dose}}^{\text{clinic}}$ as compared to the human reader ($\text{Diff}_{\text{max dose}}^{\text{clinic}}$: 3.4%, 3.5% vs 4.1%; $\text{Diff}_{\text{mean dose}}^{\text{clinic}}$: 5.0%, 5.0% vs 5.3%). However, as seen from Figure II, more OARs from the human reader have dose errors larger than 10% or 20% as compared to SOARS and SOARS-revised. Overall, new results indicate that in real clinical practice, using OAR contours from human reader, SOARS and SOARS-revised lead to generally comparable dose accuracy, while SOAR related OAR contours quantitatively have fewer number of OARs with large dose errors. We have added this new experiment, denoted as clinical dosimetric results in the revised version. Using 10 randomly selected patients to conduct this new dosimetric experiment as illustrated in Table IV and Figure II of this letter (Suppl. Table 8 and Suppl. Figure 4-5 in the revised version), the quantitative comparison results are consistent with our initial dosimetric results.

Arguably, our previous way to calculate the dosimetric error may be observed as a preliminary analysis of the dosimetric effect when directly overlying the OAR contour permutations on the “gold standard” dose maps. It eliminates the uncertainties involved in the planning step focusing on the dose difference purely brought by the OAR contour’s variation. Hence, we keep both initial (named as direct dosimetric results: $\text{Diff}_{\text{max dose}}^{\text{direct}}$ and $\text{Diff}_{\text{mean dose}}^{\text{direct}}$) and the new dose results (clinical dosimetric results $\text{Diff}_{\text{max dose}}^{\text{clinic}}$ and $\text{Diff}_{\text{mean dose}}^{\text{clinic}}$) in the revised version.

Table IV. Quantitative clinical dosimetric accuracy ($\text{Diff}_{\text{mean dose}}^{\text{clinical}}$ and $\text{Diff}_{\text{max dose}}^{\text{clinical}}$) comparison between SOARS, SOARS-revised and human reader contours on 10 randomly chosen patients from the multi-user testing dataset. Dose errors are calculated by generating new IMRT plans based on each OAR contouring permutation, and then, overly the gold standard OAR contours on top each plan. Difference in mean dose and difference in maximum dose are calculated using the Equation (8) and (9) of revised version, respectively. SOARS and SOARS-revised results are compared to human reader results, and bold values represent better performance.

OARs	Clinical dosimetric accuracy ($\text{Diff}_{\text{mean dose}}^{\text{clinical}}$ and $\text{Diff}_{\text{max dose}}^{\text{clinical}}$)					
	human reader		SOARS		SOARS-revised	
	diff in mean dose	diff in max dose	diff in mean dose	diff in max dose	diff in mean dose	diff in max dose
BrainStem	1.9% ± 1.2%	4.7% ± 7.1%	1.5% ± 0.6%	5.5% ± 4.8%	1.3% ± 1.0%	5.4% ± 5.3%
Eye_Lt	2.3% ± 1.7%	4.2% ± 2.6%	3.4% ± 2.2%	5.3% ± 3.7%	3.6% ± 2.5%	4.7% ± 2.8%
Eye_Rt	4.0% ± 2.8%	3.1% ± 3.5%	3.3% ± 3.7%	2.3% ± 2.4%	3.8% ± 3.9%	3.2% ± 2.6%
Lens_Lt	2.3% ± 2.4%	5.8% ± 6.9%	2.3% ± 2.0%	4.9% ± 5.5%	2.5% ± 2.0%	5.8% ± 5.4%
Lens_Rt	4.6% ± 4.9%	7.3% ± 7.3%	4.5% ± 4.3%	4.6% ± 6.7%	4.4% ± 3.9%	4.7% ± 6.1%
OpticChiasm	7.5% ± 9.3%	5.9% ± 8.1%	5.0% ± 5.1%	2.9% ± 3.5%	4.5% ± 4.4%	1.9% ± 1.4%
OpticNerve_Lt	13.9% ± 9.8%	5.9% ± 6.9%	11.7% ± 6.3%	2.7% ± 4.1%	11.3% ± 6.7%	2.7% ± 3.3%
OpticNerve_Rt	11.7% ± 9.1%	4.6% ± 5.5%	13.5% ± 10.1%	2.3% ± 2.0%	14.6% ± 12.2%	4.2% ± 3.4%
Parotid_Lt	5.2% ± 3.3%	3.2% ± 2.6%	4.8% ± 3.3%	2.5% ± 2.2%	4.9% ± 3.4%	2.7% ± 3.0%
Parotid_Rt	4.1% ± 3.6%	1.6% ± 1.0%	4.0% ± 2.9%	1.2% ± 0.6%	3.9% ± 3.1%	1.4% ± 1.1%
SpinalCord	6.9% ± 4.2%	6.0% ± 9.1%	5.8% ± 3.2%	4.7% ± 7.4%	5.4% ± 3.1%	3.4% ± 6.4%
TMJ_Lt	2.8% ± 2.9%	2.1% ± 1.7%	2.8% ± 2.7%	2.0% ± 2.1%	1.8% ± 1.6%	2.4% ± 2.2%
TMJ_Rt	2.0% ± 2.2%	1.7% ± 1.7%	2.4% ± 2.3%	2.5% ± 1.9%	2.9% ± 2.0%	2.6% ± 2.2%
Average	5.3%	4.1%	5.0%	3.4%	5.0%	3.5%

Figure II. The scatter plots of clinical mean dose differences ($\text{Diff}_{\text{mean dose}}^{\text{clinical}}$) and clinical maximum dose differences ($\text{Diff}_{\text{max dose}}^{\text{clinical}}$), where the new IMRT planning dose grids were generated by using the original tumor target volumes and the substitute OAR contours (SOAR, SOARS-revised, and human reader), and then, the clinical reference OAR contours were overlaid on top of each new dose grid. Blue triangle, green cross and red circle represent results of human reader, SOARS-revised and SOARS, respectively.

Q5: 162-167 (and 442-461): The “Gold” dataset is considered the baseline. Did those physicians’ contours (and/or the contours from these dosimetric datasets) get included in the training data? If so, then this is just proving that SOARS reproduces what it’s trained to reproduce. What this section is really

is a sensitivity analysis with dosimetric endpoints which are far more relevant in radiation therapy planning than DSC, HD, and ASD.

Response: The “Gold” dataset in the multi-user study (30 patients from FAH-ZU) **has not been included in the training data**. All the patients and gold standard contours of 30 patients in the multi-user dataset do not appear in the training. Training data only includes those 176 patients from CGMH (named Training-validation dataset in line 120 of original manuscript). We have made this clearer in our revision (line 195-198).

Q6: 176-177: The phrase “significantly worse” here is assumed to mean statistically significant, but it sounds like the authors are implying the magnitude of the differences are significant, when this seems not so (DICE: 69.8% vs 75.3%, HD: 8.8 vs 7.9mm, ASD: 1.6 vs 1.1mm). This should be rephrased with less hyperbole, such as: “... had inferior performance that was statistically significant ...”.

Response: Thank you for this suggestion. We agree that when we say “significantly worse”, it means statistically significant. We have rephrased using more accurate descriptions: “... had inferior performance that was statistically significant”.

Q7: 190: Has “HD95” been defined prior to this usage? HD was defined (line 171). It is assumed that HD means the maximum surface difference distance and HD95 would be the distance at which 95% of the points match ($HD95 < HD$). Both of these concepts are simple/intuitive and should be explained in detail.

Response: We apologize that it is a typo of “HD95” appeared at line 190 in the original manuscript. We evaluated all the data using the HD metric (the HD distance metrics are explained at line 415 of original manuscript). We have corrected it to “HD” in the revised version.

Q8: 207-212: Is there an observer bias when viewing existing contours and determining if they require editing, and if so, how much? I’d be interested to see a side study of physicians presented (blindly) with copies of their original contours that they drew themselves and asked if those require any editing, with the expected result being that 100% require no editing. Then, present them with contours that are different from their original opinions, where the expectation would be that 0% require no editing and 100% require minor or major edits.

Response: Thank you for this question. Following your suggestion, we have designed another blind user study to assess the observer variation/bias in evaluating the OAR editing efforts. In this blind user study, we used 30 multi-user testing patients from FAH-ZU and involved the senior physician (J. Ge) who has originally drawn the gold-standard contours of 13 OAR types in these patients. For each OAR, we randomly selected its contour from three OAR sources {gold-standard, SOARS, or the other human reader} (see Figure III in this letter and supplementary Fig 6 in the revised version) and presented it to this physician blindly. The true contour source for each OAR was kept unknown to the physician. We asked the physician to judge if each OAR contour needs editing or not. We report the number of OAR contours required for editing for each OAR source. Results are shown in the Table V of this letter and supplementary Table 9 in the revised version. From the blind user study, it is observed that 15% of the

gold-standard contours were deemed requiring further editing, which reflects the intra-observer variation on assessing the OAR revision efforts. For SOARS contours, 43% requires revision, which is slightly higher than that in the original unblind assessment by this physician where 37% SOARS contours required revision among the 13 OAR types of FAH-ZU. Since the required revision number of SOARS contours from the blind vs unblind assessment is close (43% vs. 37%), it indicates that our observer variation/bias is within an acceptably small range. Moreover, compared with SOARS, a noticeably higher number of human reader's contours require revision (55% vs 43% of SOARS), reflecting that SOARS contours' quality is generally better than the human reader's in the blind assessment. This observation is also consistent with that seen in the quantitative contouring accuracy between SOARS and the human reader (Table 5 in original manuscript). This additional analysis further strengthens our results and conclusions. We have added this new blind assessment into the revised supplementary materials.

Table V. Results of the observer variation/bias assessment in the blind user study evaluating OAR editing efforts.

	Number of OAR required editing (%)	Number of OAR without editing (%)	Total number of OAR contours assessed
Gold-standard contour	20 (15%)	110 (85%)	130
SOAR contour	54 (43%)	71 (57%)	125
Human reader's contour	74 (55%)	61 (45%)	135

Figure III. Examples of randomly selected OARs in the blind user study for the observer variation/bias assessment in evaluating the OAR editing efforts. Each OAR in a patient is randomly chosen from one of the three contouring sources {Gold, SOARS, Human-reader}. These OAR contours are presented blindly to the physician to determine if revision for any of the OARs are needed.

Q9: 226: Here you show the DICE coefficient (DSC) as a fraction ranging from 0.00 to 1.00, where elsewhere you show a percentage. I think the usual formalism is a fraction, not a percentage but regardless, choose one and be consistent throughout the paper.

Response: We apologize for using the fraction format of DSC at line 226 in the original manuscript. We have modified the paper to consistently use the percentage format for DSC.

Q10: 226: Again, the phrase “significantly improved” is a bit misleading. The changes may be statistically significant, but the absolute changes are not that large. See the prior suggestion regarding lines 176-177 and employ a similar, precise re-phrasing.

Response: Thank you for the suggestion. Same as the line 176, we have rephrased using more accurate descriptions of “statistically significantly improved”.

Q11: All these patients were from China. HN anatomy (unlike GI or pelvis) should not vary much if at all depending on nationality, but has there been any testing for other cohorts of patients outside of China? Do you expect models need to be trained based on different regions or clinical practice/protocol? If so, what is the ideal and minimal number of patient datasets required and how long does that process take?

Response: Thank you for this question. In our original version, we did include testing patients outside of China, considering that anatomies should have relatively less variation at the head and neck region between different populations as compared to the chest or abdomen. Nevertheless, in the revised version (as suggested by Reviewer 1), we added another experiment using the public HN OAR challenge dataset, i.e., MICCAI2015, where patients were collected from North America. We conducted two experiments here: (1) direct inference in the MICCAI 2015 testing cases using the CGMH pretrained SOARS model (denoted as SOARS_Inference); (2) retrain SOARS using the MICCAI 2015 training cases, then, apply the retrained SOARS model to the MICCAI 2015 testing cases (denoted as SOARS_Retrain). Results are shown in Table VI in this letter and suppl. Table 6 in the revised version. The performance of other recent leading HN OAR segmentation methods in the MICCAI 2015 dataset are also included. Note that all other methods have been retrained using the MICCAI 2015 training set except SOARS_Inference. As seen in Table VI in this letter (suppl. Table 6 in the revised version), without retraining, SOARS_Inference still achieve good performance of 80.4% mean DSC across 9 OARs, which is higher than most recent methods [2, 9-11] and only slightly lower (<1%) than the leading method UaNet [4]. This demonstrates that SOARS could handle patients from different geographic regions. If SOARS was retrained using MICCAI 2015 training cases, SOARS_Retrain further improves the performance to 83.6% DSC.

If the clinical practices (annotation protocols) are different from the international delineation guideline [14], model retraining should be carried out to adjust the deep networks so that the generated OAR contours follow the specific annotation style. We have an ongoing external clinical validation in North America on a largely non-Asian patient population (following the guideline [14]) using our trained models without modification or retraining, and the initial clinical feedback and impressions are quite positive. This may help gain more confidence on this issue. As reviewer 2 mentioned, the fact that HN anatomical structures do not vary much across different nationalities or races contributes to this performance generality as well.

From our experience, the ideal number of training patients for the OAR task is about 100~200, since OARs are normal organs requiring less training data than that in the tumor segmentation task. The

minimal number of patients depends on several factors, e.g., the size/scale of the deep model and the representativeness of selected patients. For the UNet model used in our study, the minimal training number should be at least 30. The training time also depends on several factors, such as the size of the deep model, the training patient number, the training epoch, and the speed of CPU. For ~100 patients, the training time of a single UNet model normally takes 1.5-2.5 GPU days when using the NVIDIA RTX 8000 GPU.

Table VI. Dice similarity coefficient (%) results on MICCAI2015 testing dataset. SOARS NC (direct test) refers to directly apply the pre-trained SOARS model (on CGMH) to the MICCAI 2015 testing cases, while SOARS NC (retrain) refers to first retrain SOARS using MICCAI 2015 training cases, then, apply the re-trained SOARS model on the testing cases.

	Brainstem	Mandible	Optic Chiasm	Optic Nerve		Parotid		SMG		AVG.
				left	Right	left	right	left	right	
Harrison et al [2]	87.2 ± 2.5	93.1 ± 1.8	55.6 ± 14.1	72.6 ± 4.6	71.2 ± 4.4	87.7 ± 1.8	87.8 ± 2.3	80.6 ± 5.5	80.7 ± 6.1	79.6
Tong et al. [9]	87.0 ± 3.0	93.7 ± 1.2	58.4 ± 10.3	65.3 ± 5.8	68.9 ± 4.7	83.5 ± 2.3	83.2 ± 1.4	75.5 ± 6.5	81.3 ± 6.5	77.4
AnatomyNet [10]	86.7 ± 2.0	92.5 ± 2.0	53.2 ± 15.0	72.1 ± 6.0	70.6 ± 10	88.1 ± 2.0	87.3 ± 4.0	81.4 ± 4.0	81.3 ± 4.0	79.2
FocusNet [11]	87.5 ± 2.6	93.5 ± 1.9	59.6 ± 18.1	73.5 ± 9.6	74.4 ± 7.2	86.3 ± 3.6	87.9 ± 3.1	79.8 ± 8.1	80.1 ± 6.1	80.3
UaNet [4]	87.5 ± 2.5	95.0 ± 0.8	61.5 ± 10.2	74.8 ± 7.1	72.3 ± 5.9	88.7 ± 1.9	87.5 ± 5.0	82.3 ± 5.2	81.5 ± 4.5	81.2
SOARS_Inference	87.7 ± 2.5	94.8 ± 1.6	61.8 ± 13.1	72.5 ± 8.1	72.1 ± 9.5	88.1 ± 2.5	87.7 ± 3.2	79.7 ± 7.5	79.1 ± 7.9	80.4
SOARS_Retrain	88.6 ± 2.7	96.6 ± 0.8	69.2 ± 9.8	75.8 ± 6.1	75.2 ± 4.8	88.9 ± 2.2	88.6 ± 4.8	84.5 ± 6.9	85.1 ± 5.8	83.6

Q12: Suggestion: Comment on the application of this specific method to other body sites.

Response: Our stratification method is also suitable for other body sites where anatomical structures are densely distributed with different levels of segmentation difficulty, e.g., anatomical tissues in chest or abdomen regions. We have applied this stratified method to segment 22 chest anatomical structures (lung, heart, esophagus, vessels, muscles, bones) to aid the mediastinal lymph node station parsing in a recent conference proceeding “DeepStationing: Thoracic Lymph Node Station Parsing in CT Scans using Anatomical Context Encoding and Key Organ Auto-Search, MICCAI 2021”. We now point this out in the revised version.

Q13: Question: Did you consider having professional anatomists/radiologists review, because radiation oncologists are not necessarily the best source of anatomical knowledge. (There’s a reason they vary in the quality of their contouring outputs and often draw only a subset of anatomical structures.)

Response: Thank you for this question. Diagnostic radiologists are indeed experts in anatomy. However, radiation oncologists participating in this study are also experienced in anatomical knowledge. The international HN OAR delineation guideline [14] is determined by radiation oncologists from different countries. The normal training course of radiation oncologists includes rotation to the department of diagnostic radiology during their residency and board-certified radiation oncologists would also review patients’ diagnostic images with diagnostic radiologists in weekly multispecialty head and neck cancer

tumor board meeting in the involved institutions. Nevertheless, regarding to the reviewer's question, our radiation oncologists also consulted with radiologists when encountering difficult cases, such as tumors very close to the OARs. We now mention this in the revised version.

Q14: Suggestion: Did you consider validation using multi-modality, fused CT-MR image sets where the fused MRI offers more anatomical info on soft tissue organs? Thus, the "gold standard" references could be human experts contouring on CT-MR datasets whereas SOARS is using only CT. That would be an important and significant addition to this paper.

Response: Thank you for this question and suggestion. MRI has a relatively better soft tissue resolution and is beneficial for human expert contouring. In the "gold standard" reference generation of our experiments, we indeed have included MRI images (we have made this point clearer in the revised version, e.g. line 121-123 and 386-400). As described in line 426 and 436 in the original manuscript, "Besides the planning CT scans, other clinical information, and imaging modality such as MRI (if available) were also provided to physicians as reference". So, MRIs (if available) are provided to human experts in our study when generating the "gold standard" reference contours. However, CT and MRI images are not fused together in our study. This is because hyperextension positioning under the cast fixation for CT simulation is usually used in head and neck cancer treatment, while diagnostic MR images are acquired in a neutral position. Directly fusing them using the current rigid or deformable registration algorithms often leads to large errors [15]. Hence, in our study, human experts would open two PACS windows in the computer to view CT and MRI separately to help the delineation (if they felt it necessary to consult to MRI). MRI images' availability is about 50-70% of patients in different institutions depending on each institution's practice. In contrast, SOARS is trained using only CT images (with the "gold standard" labels) and can reliably generate the OAR contours using only CT. This actually may be the strength of deep learning methods, which could use CT modality alone to achieve statistically comparable or in some scenarios better and/or more consistent performance than human experts leveraging on both CT and MRI. We have made this point clearer in the revised version (line 386-400), as we agree that it is a highly important fact to emphasize.

LIST OF MAJOR CHANGES:

1. We conducted additional experiments using two public HN OAR segmentation datasets, i.e., MICCAI 2015 and StructSeg 2019. Results are compared to the recent leading HN OAR segmentation methods.
2. We extended our multi-user testing dataset to 50 patients total (30 from FAH-ZU and 20 from SMU), and re-conducted all the experiments regarding the user studies, i.e., OAR editing effort assessment, inter-user contouring accuracy comparison, and the dosimetric accuracy comparison.
3. We conducted new clinical dosimetric evaluation experiment to evaluate the dosimetric performance by generating the new radiotherapy dose plans based on each OAR contouring permutations and then, overlaying the gold standard OARs on top of each to compare the new clinical dosimetric differences.
4. We designed new blind user study to assess the observer variation/bias in evaluating the OAR editing efforts.

5. We directly inference on the MICCAI 2015 testing cases (recruited from North America) using the CGMH pretrained SOARS model to examine SOARS performance for patients from different geographic regions.
6. We clarified that the protocol of FAH-ZU used by the human reader in the multi-user testing dataset refers to the number of OARs difference rather than the annotation difference. The human reader in the multi-user study also followed the international delineation guideline [14].
7. We clarified the data splits regarding the NAS training and the ablation study.
8. We explained the qualifications of being senior physicians in our study and clarified our method to ensure the manual delineation consensus (by following the international delineation guideline of [14]).
9. We clarified the “gold” contours in the multi-user testing dataset (from FAH-ZU) has never been included in the training stage (training is completed using only patients from CGMH).
10. We clarified the radiologist’s role in our study.
11. We clarified that MRI images (if available) were provided to physicians as reference when working on the manual delineation and explained the reasons why the direct fusion/registration of CT and MRI were not used in our study.
12. We clarified, rewrote, or rephrased all other issues raised by two Reviewers.

REFERENCES:

- [1] Guo, D. *et al.* Organ at risk segmentation for head and neck cancer using stratified learning and neural architecture search. *Proceedings of the IEEE/CVF Conference on Computer Vision and Pattern Recognition*. 4223-4232 (2020).
- [2] Harrison, A. P. *et al.* Progressive and multi-path holistically nested neural networks for pathological lung segmentation from CT images. *International conference on medical image computing and computer-assisted intervention*. 621-629 (2017).
- [3] Ronneberger, O., Fischer, P. & Brox, T. U-net: Convolutional networks for biomedical image segmentation. *International Conference on Medical image computing and computer-assisted intervention*. 234-241 (2015).
- [4] Tang, H. *et al.* Clinically applicable deep learning framework for organs at risk delineation in CT images. *Nature Machine Intelligence* **1**, 480-491 (2019).
- [5] Isensee, F., Jaeger, P. F., Kohl, S. A., Petersen, J. & Maier-Hein, K. H. nnU-Net: a self-configuring method for deep learning-based biomedical image segmentation. *Nature Methods* **18**, 203-211 (2021).
- [6] Ren, X. *et al.* Interleaved 3D-CNNs for joint segmentation of small-volume structures in head and neck CT images. *Medical Physics* **45**, 2063–2075 (2018).
- [7] Wang, Z. *et al.* Hierarchical vertex regression-based segmentation of head and neck CT images for radiotherapy planning. *IEEE Trans. Image Process.* **27**, 923–937 (2018).
- [8] Nikolov, S. *et al.* Clinically Applicable Segmentation of Head and Neck Anatomy for Radiotherapy: Deep Learning Algorithm Development and Validation Study. *J Med Internet Res* **23**, e26151, doi:10.2196/26151 (2021)

- [9] Tong, N., Gou, S., Yang, S., Ruan, D. & Sheng, K. Fully automatic multi-organ segmentation for head and neck cancer radiotherapy using shape representation model constrained fully convolutional neural networks. *Medical physics* **45**, 4558-4567 (2018).
- [10] Zhu, W. *et al.* AnatomyNet: deep learning for fast and fully automated whole-volume segmentation of head and neck anatomy. *Medical physics* **46**, 576-589 (2019).
- [11] Gao, Y. *et al.* Focusnet: Imbalanced large and small organ segmentation with an end-to-end deep neural network for head and neck ct images. *International Conference on Medical Image Computing and Computer-Assisted Intervention*. 829-838 (2019).
- [12] Liu, C., *et al.* Automatic segmentation of the prostate on CT images using deep neural networks (DNN). *International Journal of Radiation Oncology* Biology* Physics*, **104**(4), 924-932 (2019).
- [13] Lin, L. *et al.* Deep learning for automated contouring of primary tumor volumes by MRI for nasopharyngeal carcinoma. *Radiology* **291**, 677-686 (2019).
- [14] Brouwer, C. L. *et al.* CT-based delineation of organs at risk in the head and neck region: DAHANCA, EORTC, GORTEC, HKNPCSG, NCIC CTG, NCRI, NRG Oncology and TROG consensus guidelines. *Radiother Oncol* **117**, 83-90, (2015).
- [15] Head, J., *et al.* Prospective quantitative quality assurance and deformation estimation of MRI-CT image registration in simulation of head and neck radiotherapy patients. *Clinic. & transl. radiat. Oncol.* **18**: 120-127 (2019).

Reviewers' Comments:

Reviewer #1:

Remarks to the Author:

The authors clearly put an enormous effort in their revision of the manuscript. All major points from my initial review have been addressed and I have no further comments.

Reviewer #3:

Remarks to the Author:

1. Authors do not fully describe CT dataset. Scan parameters? Contrast? Etc. Would refer them to DOI: 10.1002/mp.15170 for guidelines in AI/ML reporting.

2. Authors refer to Brouwer et al for guideline OAR recommendations for contouring, but many proposed OARs were not in this paper or were made sided in the current paper but not in guidelines (eg mandible, thyroid). Please provide guideline reference for structures not in this paper, and rationale for L/R mandible and thyroid. Why separate epiglottis and GSL (epiglottis is part of the supraglottic larynx!).

3. While it may seem reasonable to compare doses to each OAR, the end goal is to obtain a similar dose distribution across the plan within a prespecified limit (eg 3%,3mm etc).

Other minor comments:

Abstract line 44/45 - 90% of OAR delineation workload; ie this work does not impact the rest of the workload

Introduction line 71: low CT contrasts - please explain further

line 72 - "enjoyed a prominent history"; please rephrase

Results line 148/149 - "ablation" - not sure what this means.

Otherwise, the authors have responded to reviewer #2.

Response to reviewers

We thank Reviewer 1 for the appreciation of "an enormous effort made in the revision" and with no further comments. We are also grateful for Reviewer 3's additional constructive comments on our manuscript. Taking them into account can further strengthen our manuscript's quality. A point-by-point response is provided below.

REVIEWER #3

Q1: Authors do not fully describe CT dataset. Scan parameters? Contrast? Etc. Would refer them to DOI: 10.1002/mp.15170 for guidelines in AI/ML reporting.

Response: Thank you very much for this suggestion. According to the AI/ML reporting guideline (DOI: 10.1002/mp.15170), we add the detailed imaging information of CT datasets from six institutions in Table I in this letter (the supplementary Table 1 in the revised version).

Table I. Detailed planning CT imaging protocols in each institution. CE represents contrast-enhanced; NC represents non-contrast.

	CGMH (n = 502)	FAH-XJU (n = 82)	FAH-ZU (n = 447)	GPH (n = 50)	HHA-FU (n = 195)	SMU (n = 227)
Scanner make	GE	Philips	Siemens	GE	Siemens	Philips
NC or CE	NC CE mixed	NC CE mixed	NC	CE	NC CE mixed	NC
Scanning parameter (voltage and current)	120kV 300mAs	120kV 280mAs	120kV 300mAs	120kV 280mAs	120kV 280mAs	120kV 275-375mAs
Spatial resolution (mm)						
Median	0.99×0.99×2.5	0.94×0.94×3.0	0.98×0.98×3.0	0.8×0.8×3.0	0.97×0.97×3.0	0.53×0.53×3.0
Minimum	0.84×0.84×1.0	0.8×0.8×1.0	0.82×0.82×1.5	0.7×0.7×3.0	0.41×0.41×1.0	0.44×0.44×3.0
Maximum	1.37×1.37×3.0	1.19×1.19×3.0	1.27×1.27×3.0	0.98×0.98×3.0	0.98×0.98×5.0	0.64×0.64×3.0

Q2: Authors refer to Brouwer et al for guideline OAR recommendations for contouring, but many proposed OARs were not in this paper or were made sided in the current paper but not in guidelines (eg mandible, thyroid). Please provide guideline reference for structures not in this paper, and rationale for L/R mandible and thyroid. Why separate epiglottis and GSL (epiglottis is part of the supraglottic larynx!).

Response: Thank you for these comments.

(1) We first clarify that mandible and thyroid are indeed included in the OAR guideline recommendations of Brouwer et al. 2015. Please see mandible described in the middle of the right column on page 2 and thyroid in the first paragraph of the right column on page 3 in Brouwer et al., 2015. Mandible is treated as separated left and right OARs because osteoradionecrosis (ORN) is a common adverse effect of **oral cavity cancer** patients receiving radiotherapy, where asymmetric radiation doses are given for left and right neck regions. Other international challenges or works also adopt left and right mandible OAR differentiation, e.g., MICCAI 2015 and StructSeg 2019 public HN OAR segmentation challenges, and reference [20-24] of the original manuscript. We would also like to analyze the relationship between received dose of thyroid and thyroid disorders, because thyroid disorders such as hypothyroidism, thyroiditis or radiation-induced thyroid cancer, are adverse effects of interest after radiotherapy.

(2) Among 42 OARs considered in our work, 37 OARs are included in the Brouwer et al. 2015 recommendation guideline [1] or references [2, 3] provided in [1] (e.g., L/R TMJ, L/R inner ear, inferior/middle/posterior constrictor muscle, etc.). We have added references [2, 3] as additional OAR guideline references in the revised version. There are indeed 5 subdivisions of brain structures not included in the Brouwer et al. 2015 recommendation guideline [1] or [2, 3], which are L/R basal ganglia, cerebellum, hypothalamus, and pineal gland. We have included these organs as OARs because radiotherapy-induced fatigue, short-term memory loss, and cognition change have been reported to be associated with the volume of scatter dose to these brain substructures [4-8]. For example, there are high risks of intracranial perineural spreading for locally advanced **NPC or nasal cavity/paranasal sinus cancer** patients. Therefore, risky nerve-bearing areas from the skull base to the cavernous sinus and intracranial fossa will be delineated within the clinical target volume (CTV) for prophylaxis or even delineated as a gross target volume (GTV) in patients with intracranial perineural spreading. Hence, in our work, patients' OARs were delineated according to both the Brouwer et al. 2015 guideline [1] (and references [2, 3] provided in [1]) and the intracranial organs PASPORT trial analyzed [4, 5]. We have added these additional references in the revised version.

(3) We apologize for making an error when using the terms epiglottis and GSL. "Epiglottis and GSL" used in our initial version of the manuscript should refer to GSL and glottic areas that Brouwer et al. 2015 recommendation guideline defined [1]. We have corrected this in Table 3 in the revised version. Thank you very much for pointing this out.

Q3: While it may seem reasonable to compare doses to each OAR, the end goal is to obtain a similar dose distribution across the plan within a prespecified limit (eg 3%, 3mm etc).

Response: Thank you for this comment. We agree that obtaining a similar dose distribution across the plan is an important end goal in clinical practice. However, the Gamma index analysis (e.g., 3%, 3mm criterion) may not be directly applicable to our scenario since it often serves as a standard quality assurance procedure to examine the passing rate within a prespecified limit, which usually requires carrying out a real radiation procedure of phantom study. In our scenario, we are examining the dosimetric impact of the auto-segmented OAR contours in the retrospective human study, where carrying out a real radiation procedure using previous patients is not feasible.

Nevertheless, following the reviewer's suggestion, we apply the Gamma index analysis in an alternative way to evaluate the overall dosimetric across the whole plan. Specifically, we treat the original clinical dose map (generated using clinical reference OAR contours) as reference dose distribution while considering the OAR-replanned dose maps (generated using substitute OAR contours, i.e., SOARS, SOARS-revised, human reader) as evaluated dose distribution. Then, for each voxel in the OAR-replanned dose map, we use the 3%/3mm rule to check if we can find a valid voxel in the original dose map. Using the validation software of PTW VeriSoft, we report the volume results from Gamma index analysis under the 3%/3mm prespecified limit. Results are shown in the following Table II of this letter. It can be observed that the volume analysis results from Gamma evaluation using different OAR contour sets, i.e., SOARS, SOARS-revised, and human reader, all achieve high values (much higher than the commonly used threshold of 95%). This may be because the target volume contours (GTV/CTV/PTV) are all fixed (since we are studying the effect of OAR contour differences). Hence, for the replanned dose

maps generated using different OAR sets, the overall dose distributions are all similar to the original dose map. This may indicate that the volume analysis results of Gamma evaluation may not be able to clearly reflect the dose impact on individual OAR when OAR contours change. Hence, for the current work, we think it is reasonable to use the mean and maximum dose metric of individual OAR (as adopted in the previous reference [9]) as the dosimetric performance analysis.

Table II. Volume results from Gamma index analysis under 3%/3mm prespecified limit when evaluated using 10 dose replanned multi-user testing patients.

	Human reader	SOARS	SOARS-revised
Patient 1	99.8%	99.8%	99.8%
Patient 2	99.6%	99.6%	99.7%
Patient 3	99.7%	99.7%	99.7%
Patient 4	99.8%	99.9%	99.8%
Patient 5	99.7%	99.7%	99.7%
Patient 6	99.7%	99.7%	99.7%
Patient 7	99.8%	99.8%	99.9%
Patient 8	99.7%	99.7%	99.7%
Patient 9	99.8%	99.8%	99.8%
Patient 10	99.9%	99.9%	99.8%
Mean	99.75%	99.76%	99.76%

Q4: Other minor comments:

Q4.1: Abstract line 44/45 - 90% of OAR delineation workload; ie this work does not impact the rest of the workload.

Response: Thank you for the question. By saying "saving 90% radiation oncologists workload" in the abstract, we refer that when edited on SOARS predictions, it would save radiation oncologists 90% contouring time (from >100 minutes to ~10 minutes).

Q4.2: Introduction line 71: low CT contrasts - please explain further.

Response: Thank you for the question. By saying "OARs often have low CT contrasts", we refer that CT intensities of some OARs are similar to their adjacent anatomic tissues and can be easily confused with adjacent tissues.

Q4.3: line 72 - "enjoyed a prominent history"; please rephrase.

Response: Thanks for pointing this out. We have rephrased to "conventional atlas-based methods have been extensively explored previously". Please see line 71-72 on page 3 in our revised manuscript.

Q4.4: Results line 148/149 - "ablation" - not sure what this means.

Response: Thanks for the question. "Ablation study" is a commonly adopted experimental evaluation used in machine learning or artificial intelligence domain, which refers to describing a procedure where

functions of certain components of the framework are examined. Hence, ablation study is used to validate the impact of each component of the framework. An introduction can be found in [https://en.wikipedia.org/wiki/Ablation_\(artificial_intelligence\)](https://en.wikipedia.org/wiki/Ablation_(artificial_intelligence)).

REFERENCES:

- [1] Brouwer, C. L. *et al.* CT-based delineation of organs at risk in the head and neck region: DAHANCA, EORTC, GORTEC, HKNPCSG, NCIC CTG, NCRI, NRG Oncology and TROG consensus guidelines. *Radiother Oncol* **117**, 83-90, (2015).
- [2] Sun, Y., *et al.* Recommendation for a contouring method and atlas of organs at risk in nasopharyngeal carcinoma patients receiving intensity-modulated radiotherapy. *Radiother Oncol* **110**, 390-397, (2014).
- [3] Christianen, M. E., *et al.* Delineation of organs at risk involved in swallowing for radiotherapy treatment planning. *Radiother Oncol* **101**, 394-402, (2011).
- [4] Nutting, C. M., *et al.* Parotid-sparing intensity modulated versus conventional radiotherapy in head and neck cancer (PARSPORT): a phase 3 multicentre randomised controlled trial. *The Lancet Oncology* **12**, 127-136, (2011).
- [5] Gulliford, S. L., *et al.* Dosimetric explanations of fatigue in head and neck radiotherapy: an analysis from the PARSPORT Phase III trial. *Radiother Oncol* **104**, 205-212, (2012).
- [6] Powell, C., *et al.* Fatigue during chemoradiotherapy for nasopharyngeal cancer and its relationship to radiation dose distribution in the brain. *Radiother Oncol* **110**, 416-421, (2014).
- [7] Kamal, M., *et al.* Fatigue following radiation therapy in nasopharyngeal cancer survivors: A dosimetric analysis incorporating patient report and observer rating. *Radiother Oncol* **133**, 35-42, (2019).
- [8] Eekers, D. B., *et al.* The posterior cerebellum, a new organ at risk?. *Clinic Translat Radiat Oncol*, 22-26, (2018).
- [9] Nelms, B. E., *et al.* Variations in the contouring of organs at risk: test case from a patient with oropharyngeal cancer. *International Journal of Radiation Oncology* Biology* Physics* **82**, 368-378, (2012).

Reviewers' Comments:

Reviewer #3:

Remarks to the Author:

Authors have mostly addressed review. I personally am not aware of any dosimetric data for partial (sided) mandible or thyroid and contouring guidelines do not recommend this breakdown.

Minor Comments:

Page 6: "delineation methods 42,43 recommended by 7." Use author name as otherwise grammatically incorrect.

Response to reviewers

We thank Reviewer 3 for the comment that “authors have mostly addressed review”. We appreciate Reviewer 3’s two minor comments and have made the accordingly changes. A point-by-point response is provided below.

REVIEWER #3

Q1: I personally am not aware of any dosimetric data for partial (sided) mandible or thyroid and contouring guidelines do not recommend this breakdown.

Response: Thank Reviewer 3 for this comment. As we explained in the previous response letter, we treat mandible as separated left and right OARs because osteoradionecrosis (ORN) can be a common adverse effect of **oral cavity cancer** patients receiving radiotherapy, where **asymmetric radiation doses** often occur for left and right neck regions. Left and right mandible OAR differentiation is also adopted by other international challenges or works, such as StructSeg 2019 HN OAR segmentation challenge and reference [20-24] in the original manuscript. A similar rationale for the separation of left and right thyroid as OARs is to analyze the relationship between the received dose of thyroid and thyroiditis.

Considering Reviewer 3’s suggestion, we choose to additionally report the segmentation results for mandible (combined left and right mandible) and thyroid (combined left and right thyroid). The segmentation results are added in the footnote of each Table in the main text and supplementary materials whenever applicable.

Q2: Page 6: "delineation methods 42,43 recommended by 7." Use author name as otherwise grammatically incorrect.

Response: Thank Reviewer 3 for pointing this out. We have corrected the citation as Reviewer 3 suggested in the final version.